# A Chronicle of Indonesia's Forest Management: A Long Step towards Environmental Sustainability and Community Welfare

Hunggul Yudono Setio Hadi Nugroho [1,*], Yonky Indrajaya [1], Satria Astana [2], Murniati [1], Sri Suharti [1], Tyas Mutiara Basuki [1], Tri Wira Yuwati [1], Pamungkas Buana Putra [1], Budi Hadi Narendra [1], Luthfy Abdulah [1], Titiek Setyawati [1], Subarudi [3], Haruni Krisnawati [1], Purwanto [1], M. Hadi Saputra [1], Yunita Lisnawati [1], Raden Garsetiasih [4], Reny Sawitri [1], Indra Ardie Surya Liannawatty Purnamawan Putri [1], Ogi Setiawan [1], Dona Octavia [1], Hesti Lestari Tata [1], Endang Savitri [1], Abdurachman [1], Acep Akbar [1], Achmad Rizal Hak Bisjoe [1], Adi Susilo [1], Aditya Hani [1], Agung Budi Supangat [1], Agung Wahyu Nugroho [1], Agus Kurniawan [1], Ahmad Junaedi [1], Andhika Silva Yunianto [1], Anita Rianti [1], Ardiyanto Wahyu Nugroho [1], Asep Sukmana [1], Bambang Tejo Premono [1], Bastoni [1], Bina Swasta Sitepu [1], Bondan Winarno [1], Catur Budi Wiati [1], Chairil Anwar Siregar [1], Darwo [1], Diah Auliyani [1], Diah Irawati Dwi Arini [4], Dian Pratiwi [1], Dila Swestiani [1], Donny Wicaksono [1], Dony Rachmanadi [1], Eko Pujiono [1], Endang Karlina [1], Enny Widyati [1], Etik Erna Wati Hadi [1], Firda Mafthukhakh Hilmya Nada [1], Fajri Ansari [1], Fatahul Azwar [1], Gerson Ndawa Njurumana [1], Hariany Siappa [1], Hendra Gunawan [1], Hengki Siahaan [1], Henti Hendalastuti Rachmat [1], Heru Dwi Riyanto [1], Hery Kurniawan [1], Ika Heriansyah [1], Irma Yeny [1], Julianus Kinho [4], Karmilasanti [1], Kayat [4], Luthfan Meilana Nugraha [1], Luthfi Hanindityasari [1], Mariana Takandjandji [4], Markus Kudeng Sallata [1], Mawazin [1], Merryana Kiding Allo [1], Mira Yulianti [1], Mohamad Siarudin [1], Muhamad Yusup Hidayat [1], Muhammad Abdul Qirom [1], Mukhlisi [4], Nardy Noerman Najib [1], Nida Humaida [1], Niken Sakuntaladewi [1], Nina Mindawati [1], Nining Wahyuningrum [1], Nunung Puji Nugroho [1], Nur Muhamad Heriyanto [1], Nuralamin [1], Nurhaedah Muin [1], Nurul Silva Lestari [1], Oki Hidayat [4], Parlin Hotmartua Putra Pasaribu [1], Pratiwi [1], Purwanto [1], Purwanto Budi Santosa [1], Rahardyan Nugroho Adi [1], Ramawati [1], Ratri Ma'rifatun Nisaa [1], Reni Setyo Wahyuningtyas [1], Resti Ura [1], Ridwan Fauzi [1], Rosita Dewi [1], Rozza Tri Kwatrina [1], Ryke Nandini [1], Said Fahmi [1], Sigit Andy Cahyono [1], Sri Lestari [1], Suhartono [1], Sulistya Ekawati [5], Susana Yuni Indriyanti [1], Tien Wahyuni [1], Titi Kalima [1], Tri Atmoko [4], Tri Rizkiana Yusnikusumah [1], Virni Budi Arifanti [1], Vivi Yuskianti [1], Vivin Silvaliandra Sihombing [1], Wahyu Catur Adinugroho [1], Wahyudi Isnan [1], Wanda Kuswanda [4], Wawan Halwany [1], Wieke Herningtyas [1], Wuri Handayani [1], Yayan Hadiyan [1] and Yulizar Ihrami Rahmila [1]

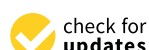



[1] Research Center for Ecology and Ethnobiology, National Research and Innovation Agency (BRIN), Jalan Raya Jakarta-Bogor Km 46, Cibinong 16911, Indonesia; yonk001@brin.go.id (Y.I.); murn006@brin.go.id (M.); sris021@brin.go.id (S.S.); tyas008@brin.go.id (T.M.B.); triw016@brin.go.id (T.W.Y.); rpam001@brin.go.id (P.B.P.); budi065@brin.go.id (B.H.N.); lutf008@brin.go.id (L.A.); titi025@brin.go.id (T.S.); haru005@brin.go.id (H.K.); purw022@brin.go.id (P.); mhad002@brin.go.id (M.H.S.); yuni028@brin.go.id (Y.L.); reny004@brin.go.id (R.S.); indr055@brin.go.id (I.A.S.L.P.P.); ogis001@brin.go.id (O.S.); dona002@brin.go.id (D.O.); made009@brin.go.id (H.L.T.); enda052@brin.go.id (E.S.); abdu082@brin.go.id (A.); acep004@brin.go.id (A.A.); achm046@brin.go.id (A.R.H.B.); adis013@brin.go.id (A.S.); adit029@brin.go.id (A.H.); agun037@brin.go.id (A.B.S.); agun038@brin.go.id (A.W.N.); agus159@brin.go.id (A.K.); ahma112@brin.go.id (A.J.); andh008@brin.go.id (A.S.Y.); anit014@brin.go.id (A.R.); ardi015@brin.go.id (A.W.N.); asep057@brin.go.id (A.S.); bamb061@brin.go.id (B.T.P.); bast001@brin.go.id (B.); bina004@brin.go.id (B.S.S.); bond005@brin.go.id (B.W.); catu007@brin.go.id (C.B.W.); chai008@brin.go.id (C.A.S.); darw004@brin.go.id (D.); diah015@brin.go.id (D.A.); dian078@brin.go.id (D.P.); dila001@brin.go.id (D.S.); donn013@brin.go.id (D.W.); dony004@brin.go.id (D.R.); ekop010@brin.go.id (E.P.); enda050@brin.go.id (E.K.); enny008@brin.go.id (E.W.); etik004@brin.go.id (E.E.W.H.); fmaf001@brin.go.id (F.M.H.N.); fajr005@brin.go.id (F.A.); fata004@brin.go.id (F.A.); gers001@brin.go.id (G.N.N.); hari054@brin.go.id (H.S.); hend052@brin.go.id (H.G.); heng003@brin.go.id (H.S.); hent003@brin.go.id (H.H.R.); heru026@brin.go.id (H.D.R.); hery011@brin.go.id (H.K.);

ikah001@brin.go.id (I.H.); irma015@brin.go.id (I.Y.); karm003@brin.go.id (K.); luth007@brin.go.id (L.M.N.); luth008@brin.go.id (L.H.); mark001@brin.go.id (M.K.S.); mawa001@brin.go.id (M.); merr004@brin.go.id (M.K.A.); mira011@brin.go.id (M.Y.); moha073@brin.go.id (M.S.); muha284@brin.go.id (M.Y.H.); muha283@brin.go.id (M.A.Q.); nard001@brin.go.id (N.N.N.); nida001@brin.go.id (N.H.); nike009@brin.go.id (N.S.); nina017@brin.go.id (N.M.); nini011@brin.go.id (N.W.); nunu005@brin.go.id (N.P.N.); nurm012@brin.go.id (N.M.H.); nura034@brin.go.id (N.); nurh037@brin.go.id (N.M.); nuru042@brin.go.id (N.S.L.); parl001@brin.go.id (P.H.P.P.); prat007@brin.go.id (P.); purw023@brin.go.id (P.); purw021@brin.go.id (P.B.S.); raha014@brin.go.id (R.N.A.); rama009@brin.go.id (R.); ratr002@brin.go.id (R.M.N.); reni009@brin.go.id (R.S.W.); rest016@brin.go.id (R.U.); ridw010@brin.go.id (R.F.); rosi015@brin.go.id (R.D.); rozz001@brin.go.id (R.T.K.); ryke001@brin.go.id (R.N.); said003@brin.go.id (S.F.); sand009@brin.go.id (S.A.C.); sril002@brin.go.id (S.L.); suha038@brin.go.id (S.); susa010@brin.go.id (S.Y.I.); tien003@brin.go.id (T.W.); titi028@brin.go.id (T.K.); trir005@brin.go.id (T.R.Y.); virn002@brin.go.id (V.B.A.); vivi005@brin.go.id (V.Y.); vivi006@brin.go.id (V.S.S.); wahy060@brin.go.id (W.C.A.); wahy061@brin.go.id (W.I.); wawa022@brin.go.id (W.H.); wiek001@brin.go.id (W.H.); wuri003@brin.go.id (W.H.); yaya027@brin.go.id (Y.H.); yuli054@brin.go.id (Y.I.R.)

2   Research Centre for Behavioral and Circular Economics, National Research and Innovation Agency (BRIN), Jalan Gatot Subroto No. 10, Jakarta 12710, Indonesia; satr013@brin.go.id

3   Research Center for Population, National Research and Innovation Agency (BRIN), JL. Gatot Subroto No. 10, Jakarta 12710, Indonesia; suba010@brin.go.id

4   Research Center for Applied Zoology, National Research and Innovation Agency (BRIN), Jalan Raya Jakarta-Bogor Km 46, Cibinong 16911, Indonesia; rade040@brin.go.id (R.G.); diah014@brin.go.id (D.I.D.A.); juli008@brin.go.id (J.K.); kaya001@brin.go.id (K.); mari049@brin.go.id (M.T.); mukh014@brin.go.id (M.); okih001@brin.go.id (O.H.); tria019@brin.go.id (T.A.); wand002@brin.go.id (W.K.)

5   Research Centre for Society and Culture, National Research and Innovation Agency (BRIN), Jl. Gatot Subroto No. 10, Jakarta 12710, Indonesia; suli018@brin.go.id

*   Correspondence: hunggul.yudono.setio.hadinugroho@brin.go.id

**Abstract:** Indonesia is the largest archipelagic country in the world, with 17,000 islands of varying sizes and elevations, from lowlands to very high mountains, stretching more than 5000 km eastward from Sabang in Aceh to Merauke in Papua. Although occupying only 1.3% of the world's land area, Indonesia possesses the third-largest rainforest and the second-highest level of biodiversity, with very high species diversity and endemism. However, during the last two decades, Indonesia has been known as a country with a high level of deforestation, a producer of smoke from burning forests and land, and a producer of carbon emissions. The aim of this paper is to review the environmental history and the long process of Indonesian forest management towards achieving environmental sustainability and community welfare. To do this, we analyze the milestones of Indonesian forest management history, present and future challenges, and provide strategic recommendations toward a viable Sustainable Forest Management (SFM) system. Our review showed that the history of forestry management in Indonesia has evolved through a long process, especially related to contestation over the control of natural resources and supporting policies and regulations. During the process, many efforts have been applied to reduce the deforestation rate, such as a moratorium on permitting primary natural forest and peat land, land rehabilitation and soil conservation, environmental protection, and other significant regulations. Therefore, these efforts should be maintained and improved continuously in the future due to their significant positive impacts on a variety of forest areas toward the achievement of viable SFM. Finally, we conclude that the Indonesian government has struggled to formulate sustainable forest management policies that balance economic, ecological, and social needs, among others, through developing and implementing social forestry instruments, developing and implementing human resource capacity, increasing community literacy, strengthening forest governance by eliminating ambiguity and overlapping regulations, simplification of bureaucracy, revitalization of traditional wisdom, and fair law enforcement.

**Keywords:** sustainable forest management; policy dynamics; shifting paradigm

## 1. Introduction

Indonesia is the world's largest archipelagic country, where 63% of its total land area, amounting to 120.5 million hectares, is designated as a State Forest Area [1]. Although occupying only 1.3% of the world's land area, Indonesia possesses the third-largest rainforest and the second-highest level of biodiversity, with very high species diversity and endemism [2]. The long history of managing forest areas in Indonesia is marked by policy changes from before independence to post-reform. Besides being known as a country with a wealth of forest resources, during the last two decades, Indonesia has also become known as a country with a high level of deforestation, a producer of smoke from burning forests, and a producer of carbon emissions. The government of Indonesia is committed to reducing deforestation and carbon emissions while enhancing community welfare [3].

Along with policy changes in forest management, the rate of deforestation continued to decline [4]. Deforestation was 115.46 thousand ha year$^{-1}$ in the 2019–2020 period, which is the lowest deforestation rate in history [5]. This value is much lower than the deforestation in 2018–2019 of 462.46 thousand ha [4]. The decline in the rate of deforestation is also in line with the decrease in the area of degraded land. After consistently increasing from 1974 to 2004, the area of degraded land has consistently decreased until the latest data for 2018 [6].

Important steps taken by the Indonesian government include a moratorium on permits for the use of primary natural forests and peatlands, land rehabilitation and soil conservation, environmental protection, and other important regulations, which have resulted in various important policy implications that have led to a reduction in deforestation and land degradation. The challenge that still exists is the division of administrative areas and population growth, which have implications for the conversion of forest land to non-forest land. Many rural residents live close to the forest and depend on it for their livelihood. From the 2018 data, out of a total of 83,931 villages in Indonesia, 2768 villages (3.30%) are located in forest areas, and 18,617 villages (22.18%) are located on the edges of forest areas [7]. In general, they belong to a group of financially deprived people whose lives are very dependent on forests.

Technically, the challenge of sustainable forest management can be formulated into three main issues: (i) stopping the rate of destruction of forests and their ecosystems and restoring those that have already been damaged, (ii) developing natural wealth to increase people's prosperity, and (iii) encouraging the participation of communities around forests in maintaining and preserving forest functions.

This paper reviews the long steps of Indonesian forest management toward achieving environmental sustainability and community welfare. The aim of this paper is to review the environmental history and the long process of Indonesian forest management towards achieving environmental sustainability and community welfare. To do this, we analyze the milestones of Indonesian forest management history, existing policy and strategy, present and future challenges, and provide strategic recommendations toward a viable Sustainable Forest Management (SFM) system. The sources for this paper consist of scientific publications, research reports, and other relevant materials, as well as the long experience of the authors as researchers at the Ministry of Environment and Forestry from the start of the establishment of the R&D institute at the Ministry of Environment and Forestry until the merging of all R&D institutions into the National Research and Innovation Agency (Badan Riset dan Inovasi Nasional/BRIN).

## 2. Milestones in Indonesian Forestry History

This section examines the historical evolution of forest management in Indonesia, which is a contested field with regard to who holds the right to access, manage, own, and control land and forest resources. As well as discusses the dynamics in domestic policies, laws, and regulations at the national and regional levels that influence the development of forest management in Indonesia.

### 2.1. Socio-Political Dynamics

Forest management, which mostly focuses on the issue of environmental conservation and socio-political dynamics, is influenced by social attitudes toward "going green" and political policies involving both social and political factors. Politics itself has two sides: in a good sense, it is "a noble quest for good order and social justice", while at its worst, politics is "a selfish grab for power, glory, and riches" [8]. Most of the discussion in this section focuses especially on the dominance of power that causes conflict in forest management in Indonesia, which can be categorized into three periods: before and during the Dutch and Japanese colonial periods, post-independence forest management: from the Old Order to the New Order; and the Reformation period—now (Figure 1) [9,10].

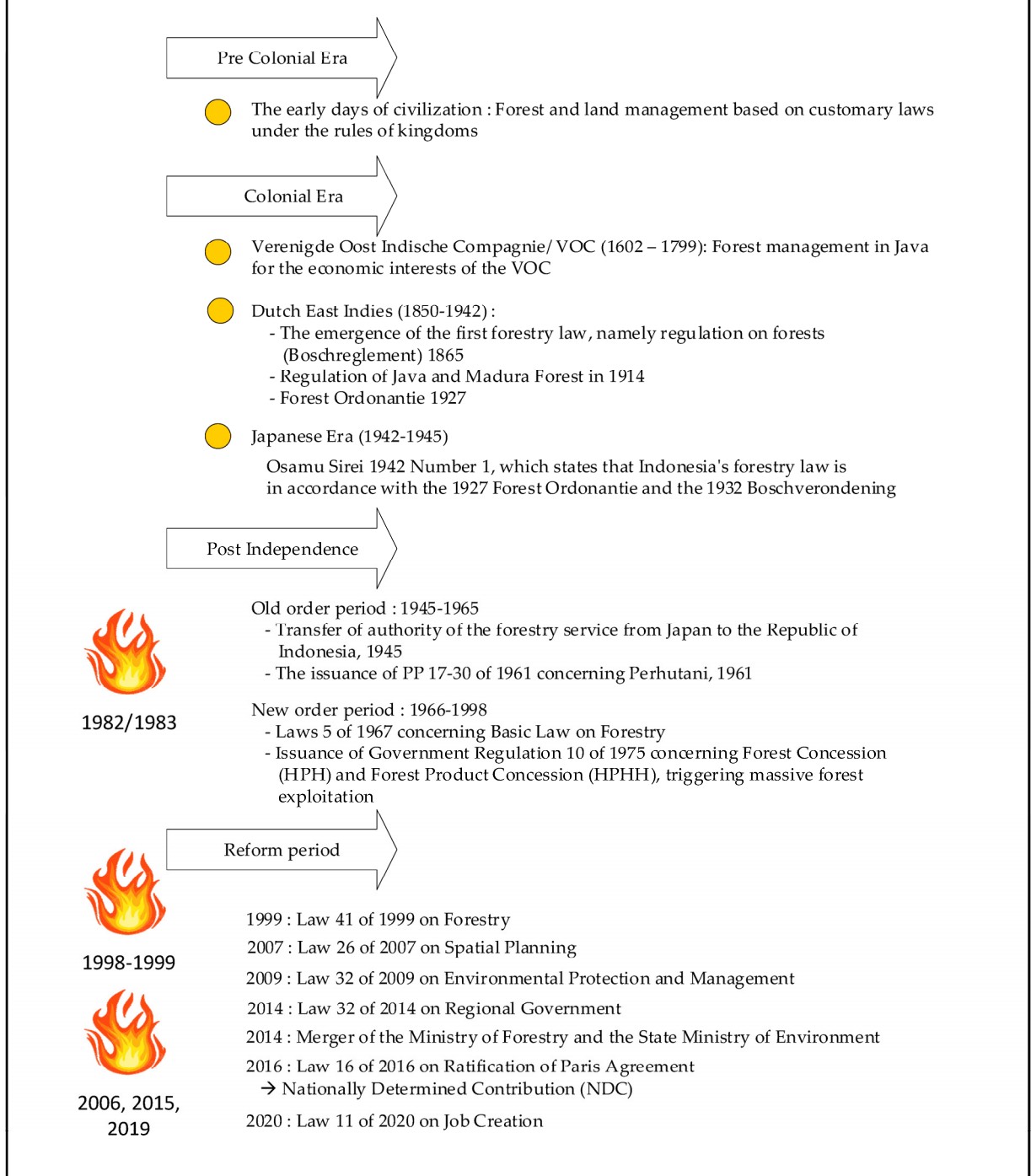

**Figure 1.** The dominance of historic and current political power in Indonesian forest management.

### 2.1.1. Pre-Colonial until the Dutch and Japanese Colonial Period

The initial study on the socio-political dynamics of forest management in Indonesia began in the pre-colonial period, when the indigenous people managed the natural resources, including forests, which were regulated by the customary law of each customary community. The dominance of customary law is strong as a reference in determining social relations between community members and also regulating how communities manage forests. In this context, conflicts are resolved by customary law [11]. Other studies reported that the dynamics of forest management in Indonesia began when the country was still composed of small and large kingdoms. At that time, those who enjoyed the most agrarian products were the king and the royal family, and the forest was a symbol of glory and power for the people. The interests of the people seem to be underestimated, and the interests of the king are prioritized, resulting in many social conflicts because the kingdom is not rooted in the interests of the people [10,12,13]. Riley [14] stated that dominance occurred as it was supported by two pillars, i.e., the physical pillar, where physical strength was used against groups of minorities of dissidents, and the ideological pillar, where hegemony through philosophy, culture, and ideology was used to obtain approval from others.

During the colonial period until 1945, the dominance of power in natural resource management was held by the Dutch and Japanese colonial governments. During this period, deforestation occurred due to policies that allowed forest clearing for construction, shipbuilding, and land clearing permits for agricultural purposes, which converted natural forests into sugarcane, coffee, indigo, and rubber plantations. Conflicts related to forest management were resolved following the regulations of the Dutch and Japanese colonial governments. This is confirmed by Reynolds et al. [15], who found that those in dominant positions—in this case, colonials—use them to create and maintain perceptions of legitimacy and stability.

### 2.1.2. Post-Independence: From the Old Order to the New Order

After Indonesia proclaimed its independence on 17 August 1945, the responsibility of forest management shifted from the Dutch and Japanese to the Indonesian government. The government began to organize legal arrangements for forest management to adapt to the conditions of Indonesia as an independent country. The adaptation was carried out by translating the forestry laws produced during the colonial period to conform with the Preamble of the 1945 Constitution. In addition to this, work guidelines were also prepared by the Forestry Service to outline Indonesian government policies regarding forest management. However, forest management laws were still in the form of forestry laws and regulations inherited from the Dutch colonial government [9,10].

There was a political upheaval in 1965 that changed the government system. The New Order government replaced the Old Order; during this period, forestry management matters came under the authority of the Ministry of Agriculture. To pursue economic growth, policies were issued that attracted domestic and foreign investors [9,16]. Foreign investors flocked to Indonesia. As a result, Indonesia's forest resources are massively exploited by granting forest concessions, especially in Sumatra, Kalimantan, Sulawesi, Maluku, and Irian Jaya (recent Papua). The granting of concession rights not only makes a positive contribution but also has a negative impact on increasing Indonesia's economic growth due to practices of corruption, collusion, and nepotism that have taken the "development victims" out of local communities [17,18].

### 2.1.3. Reform Period after 1998–Now

There was a massive reform movement that reached its peak in 1998 to end the New Order, but the legal phenomenon of forest resource management did not change ideologically and did not reflect the spirit and aspirations of the reform movement. Even though there are many demands to formulate policies that are more directed towards saving forests than achieving economic development targets, the government's dominance in policy formulation and decision-making is still very strong [18,19]. Swyngedouw [20]

stated that government dominance has several grounds, including the idea that few people know best, giving them the confidence to make political decisions when political problems are framed in technological/managerial issues so that technical experts or managers make decisions based on exclusive skills.

Meanwhile, at the global and national levels, there has been a shift in the paradigm of forest management policies in which community participation is the focus of attention and utilization of all the resources contained therein for the prosperity of the people [21]. In response, the Ministry of Forestry established a policy called Community-Based Forest Management (CBFM) in 1995 [10,21,22]. The change reflects the awareness that forest management in Indonesia will not work if it only considers the benefit principle (the anthropocentric paradigm). Community-Based Forest Management (CBFM), or currently better known as Social Forestry (SF), is experiencing dynamic development in accordance with the complexity of forest management problems facing global demands [23–25].

### 2.2. Forest Policies, Laws, and Regulation in Indonesia

In Indonesia, all land is legally classified into forest areas and non-forest areas. Forest areas are managed by both state-owned companies and local governments. However, the actual condition in the field is that many non-forested areas are forested, and vice versa. It was identified that 26 million ha of forest estate were highly degraded, with little or no vegetation left [26]. This section will describe the dynamics of forestry policies, laws, and regulations related to forest land use in Indonesia. Spatial policy conflicts occur, which are generally caused by differences in reference between the spatial plans stipulated by the central government and regional regulations concerning provincial spatial plans (PSP/Provincial Spatial Plans/RTRWP).

### 2.2.1. Forest Policy in Indonesia

The basic forest policy in Indonesia is rooted in Article 33 of the 1945 Constitution and reflected in Forest Law No. 5/1967 [27]. There are three important phases in the spatial planning of forest areas in Indonesia: (1) Forest Land Use and Solidification Plan (RPHH)/Forest Land Use by Consensus (TGHK), (2) Provincial Spatial Plan (PSP/RTRWP), and (3) Synchronization of TGHK and RTRWP [28].

1. Forest Land Use and Solidification Plan (RPPH)/Forest Land Use by Consensus (TGHK)

Spatial management efforts in Indonesia have started since the Fifth Year Development Plan I (1969–1974) with various spatial plans. RPPH, known as for provinces outside of Java Island, has been developed in the forestry industry and encompasses maps for every province in Indonesia. TGHK is developed as an elaboration of Law No. 5 of 1967 concerning the Main Provisions of Forestry, Government Regulation No. 33 of 1970 about Forest Planning, and several Decrees of the Minister of Agriculture. Based on TGHK, the forest area in Indonesia is 141,774,427 ha (Statistics of Indonesian Forestry 1990/1991).

The use of TGHK as a reference has implications for juridical, biophysical, and technical as well as socio-economic development in regions, sectors, and communities. Problems arose because, at the time the TGHK was being prepared, the need for land use by institutions other than the Forestry Service had not become an urgent need. Furthermore, the TGKH was not supported by data/maps related to forest land cover or land/forest maps that were claimed to belong to the community.

2. Provincial Spatial Plan (PSP/RTRWP)

Law No. 24 of 1992, which was revised by Law No. 26 of 2007, was issued to increase the efficiency of spatial utilization and harmony in regional growth and social welfare. This law aims to integrate various spatial arrangements that are sector-oriented into one unitary, mutually integrated, and functional environment. Based on this law, all provinces are obliged to prepare RTRWP, which represents the strategy and structure of spatial utilization in the provincial region and contains guidance for the management of the protected area

and cultivation area, urban and rural areas, as well as specific areas, for 15 years and is legislated with Regional Regulation.

3. Synchronization of TGHK and RTRWP

Based on awareness of the need to synchronize the TGHK map and RTRWP map in all provinces, the Spatial Study Team of the Ministry of Forestry was established with the Decree of Minister of Forestry No. 726/Kpts-II/93. The activities of this Ministry of Forestry Study Team were called 'Synchronization of TGHK and RTRWP'. Over time, these activities received political support with the issuance of (1) Instruction of Minister of Home Affairs No. 474/4263/Sj to all Governors in Indonesia to carry out synchronization of TGHK and RTRWP, and the result was used as material to review RTRWP; (2) Decree of State Minister for National Development Planning/Chairman of BAPPENAS No.KEP.001/KET/1/1995 about the Technical Team of National Spatial Use Management, which included the activities of synchronizing TGHK and RTRWP as one of the Tasks of Technical Team II coordinated by the Ministry of Home Affairs (MoHA). The forest area based on the result of the synchronization of TGHK and RTRWP was 120,353,104 ha.

### 2.2.2. Forest Laws and Regulations in Indonesia

In Indonesia, managing forestland has three basic goals: fostering economic growth, enhancing rural livelihoods, eradicating poverty, and generating environmental services and benefits [29]. Indonesia's Forestry Law No. 41/1999 declares that all forested areas in Indonesia without private rights are considered state forelands. Moreover, to support the production, protection, and conservation functions of forests, the Ministry of Forestry (MoF) designed integrated forestland use zoning in 1970 [30]. A detailed description of key forest regulations in Indonesia and their implications for forest management can be seen in Table A1 (Appendix A).

### 3. Forest Resources Management: Existing Policies and Strategy

*3.1. Forestry Planning and Environmental Management*

Forestry planning is part of forest management and is intended to provide guidelines and directions that ensure the achievement of the objectives of forestry administration, namely sustainable forest management for the greatest prosperity of the people in a just and sustainable manner.

### 3.1.1. Forest Inventory

Forest inventory is a series of activities to find out and obtain complete data and information on the forest's resources, potential natural resources, and environment. The implementation is conducted at the national, provincial, watershed, and forest management units (FMU), each of which is carried out at least once every five years [31]. The data and information collected in the activities include (a) forest cover, (b) types and potential of forest stands, and (c) types, potentials, and distribution of non-timber plants or hydrological aspects at the watershed level. The results are used as the basis for the inauguration of forest areas, the forest resource balance sheet, forestry plans, forest information systems, and policy formulation [32]. The forest inventories are also used for greenhouse gas (GHG) calculations in the forestry sector (stock, absorption, and carbon emissions), as well as a national peat ecosystem inventory and mapping the potential for biodiversity [1].

The government's limitations in providing resources for inventory implementation should be overcome by involving the private sector, NGOs, and community participation. More accurate information on potential forest resources will be produced by optimizing the use of high-resolution satellite images in monitoring forest resources. For peat ecosystems, an inventory on a larger scale has also been carried out and produced a map of the hydrological area of peat and a map of the function of the peat ecosystem at a scale of 1:50,000 [1]. The inventory results are presented in the National Forest Monitoring System. This system provides spatially based data and information on forest resources to inform the public of the dynamics of current forest conditions quickly and accurately. This system was

built with data sources from monitoring forest cover conditions based on remote sensing technology and a national forest inventory system based on terrestrial measurements.

### 3.1.2. Forest Area Gazettement

Forest area gazettement aims to achieve legal certainty regarding the status, boundaries, and area of the forest area. Its activities include the stages of designation/appointment of forest areas, boundary demarcation, and determination of forest areas [33]. The process of forest gazettement has come a long way, is not easy, and was initiated during the Dutch colonial period [34]. In 1950, the area of forest that had been gazetted was 16.8 million hectares [35]. Of the total forest area (125.8 million ha) in 2021, the forest area of 89.9 million ha, or 71.43%, has been determined [1]. The process of gazettement of forest areas in Indonesia is generally constrained in the face of overlapping uses and community property rights claims, as well as limited resources, compared with the area of forest areas that have not been gazetted [36,37]. The government targets the completion of the gazettement of forest areas in 2024 by including this activity in a national strategic project. Another acceleration strategy is the active involvement of local communities, especially in boundary demarcation stages [38]. In addition to the economic benefits, local community participation also helps to ensure the boundaries between private land and forest areas. This is important because legal and legitimate status must also follow fast mass boundary mapping [34].

### 3.1.3. Forest Use Management

Based on the forestry law, the use of forest areas can be carried out in all forest functions by taking into account their nature, main function, characteristics of use, and vulnerability. Forest areas that cannot be utilized are only in strict nature reserves, core zones, and forest zones in national parks [39]. Most of the forest area concessions are in production forests, with an allocation of 34.18 million hectares, of which 55% have been granted forest business licenses. There are several types of forest business licenses: business licenses for the utilization of timber forest products from natural forests, business licenses for the utilization of timber forest products from industrial plantation forests, business licenses for the utilization of timber forest products from ecosystem restoration, licenses for collecting non-timber forest products, environmental services business licenses, and social forestry schemes [1].

The granting of business licenses to entrepreneurs in large numbers, which began intensively in the early 1970s, has created injustice for the community because there is so little legal access to forest area management. This condition also triggers tenurial conflicts and illegal land grabbing. The government is responding to this by promoting agrarian reform policies that are focused on the process of allocation and consolidation of ownership, access, and use of forest areas through the Social Forestry (PS) and Lands for Agrarian Reform (TORA) programs [40].

Until July 2022, the forest area used for social forestry programs reached 5 million ha. Distribution was granted to 1,106,221 families spread across 33 provinces and 367 districts. Social forestry management was established through village forests, community forests, community plantation forests, private forests, adat forests, and forestry partnerships [1].

To provide legal certainty over land tenure by communities in forest areas and resolve conflicts in forest areas, the government is implementing Lands for Agrarian Reform (TORA) by means of land redistribution and asset legalization. Until 2020, TORA's achievements are around 2.66 million hectares, or about 55% of the 5th revised TORA indicative map area, covering an area of 4.85 million hectares [41]. This achievement must be improved by overcoming obstacles in the field, such as the lack of synergy and coordination among stakeholders at the central and regional government levels [42]. Socialization of the TORA program and capacity building should also be encouraged so that stakeholders and the community have the same understanding of rights and responsibilities in implementing this scheme [43].

Recently, the Ministry of Environment and Forestry made a new policy on forest areas with special management (KHDPK) on Java Island through Minister of Environment and Forestry Regulation No. 7/2021. As a result, forest areas that Perum Perhutani has managed since the Dutch era will be relocated. KHDPK has nominated six types of management: social forestry, utilization of environmental services, use of forest areas, structuring of forest areas in the context of forest area gazettement, rehabilitation, and forest protection. This transition mode is necessary because institutional changes in function are always slower than physical changes in the forest. The physical field always requires the presence of forest managers. If Perhutani and the government are late, their functions will be replaced by free riders.

### 3.1.4. Establishment of the Forest Management Area

To realize efficient and sustainable forest management, the government has launched a forest management area outside Java Island at the provincial level and below, which consists of a collection of Forest Management Units (FMU). Based on the dominant function of the forest area, the unit can be in the form of production forest management units (KPHP), protection forest management units (KPHL), or conservation forest management units (KPHK) [44].

FMU development can link forest management and tenure conflict resolution at the site level through spatial management and community management rights to natural resources, including community partnerships with permit holders [45]. There are at least three advantages to developing FMUs for promoting good forest governance [46]: (a) resolution of resource conflicts; (b) efficiency of management; and (c) facilitating local socio-economic institutions.

However, in its development, the existence of FMUs has faced several challenges, mainly related to regional autonomy policies and authority in the forestry sector, particularly the management of protected forests and production forests under the regional Forestry Service [47]. In this condition, the Forestry Service can be encouraged to carry out administrative functions in the future, while FMUs will be in charge of forest management functions. The Forestry Service is more positioned as an agency that plans and produces forestry policies in its area, while the FMU is fully responsible for operational activities based on policy guidelines from the Forestry Service [48].

### 3.1.5. Forestry Planning

On a national scale, the preparation of forestry plans includes forest area plans and forestry development plans based on area functions, planning period, and geographic scale [32]. However, its implementation has several obstacles, such as not yet integrating spatial planning with a forest landscape approach, including watersheds, biodiversity, and communities. Another problem is the dualism of forest and agricultural spatial systems, causing uncertainty in land use planning and forestry conflicts both horizontally (between government agencies) and vertically (between forestry authorities and forestry companies or local communities) [49–51]. Integration is important as the mainstream of forest planning in Indonesia to assist decision-makers in considering multi-purpose functions and environmental challenges [52]. In addition, good coordination between institutions is also needed in sustainable forest management when formulating laws on forest-based spatial planning, and there are appropriate mechanisms and tools for formulating sustainable forest management technologies [53].

### 3.2. Sustainable Production Forest Management

The exploitation of wood from natural forests is still a topic of debate regarding economic and ecological issues. The debate regarding the management of production forests to produce timber is often linked to the acquisition of unequal economic value and the impact of the damage it causes. For this reason, policymakers need to conduct an appropriate valuation of the impact of harvesting on remaining standing, site conditions,

wildlife, and post-harvest land fragmentation [54]. The valuation in question is not only the ecosystem value that will be sacrificed but also the overall impact, considering the very varied types of ecosystems and the biodiversity content in them. The application of the selective logging silvicultural system still needs to be tested further, especially on the five aspects of the assessment, namely: (1) related to the impact of damage that will occur according to type, type of forest or landscape, (2) related to forest processes for self-repair or adaptation to the damage caused, (3) ecosystem response to damage is sometimes non-linear and even above the threshold of ecological processes, species interactions, and population sizes, (4) the need for certain taxa in the association as a unified natural system, and (5) changes such as logged-over forests that are flammable, opening access for the use of remaining trees, or even the appearance of exotic species after harvesting. For this reason, Lindenmayer and Laurance [54] suggest that the management of production forests should carefully consider the ecosystem type. This is because sometimes the highest biodiversity does not consider topography. In addition, the duration of the utilization permit is only based on economic feasibility without considering the ability of the tree to grow naturally as a forest stand.

*3.3. Natural Resources and Ecosystem Conservation*

The legal umbrella for the conservation of natural resources and ecosystems in Indonesia is Law Number 5 of 1990 and Law Number 41 of 1999, which regulate conservation areas and biodiversity. Indonesia has adopted three world conservation strategies as pillars of conservation management: maintenance of essential ecological processes and life-support systems, preservation of genetic diversity, and sustainable utilization of species and ecosystems [55].

In 1993, Indonesia developed a guideline for managing biodiversity entitled Biodiversity Action Plan for Indonesia. This is further improved and updated to become the Indonesian Biodiversity Strategy and Action Plan (IBSAP) for 2003–2020. For this, Indonesia has tried to address the issue of meeting optimum utilization and sustainably managing biodiversity. The first period of IBSAP is 2003–2020 and is periodically updated (the last update was from 2015 to 2020). This document also guides the implementation of biodiversity management and conservation in line with global targets and directions as mandated under Decision X/2, COP 10 UNCBS, Nagoya, and supports the UNCBD strategic plan. This will be a legally binding document that oversees the implementation of Act Number 5 in the year 1994 (ratification of UNCBD).

Indonesia has allocated forest areas that function for the conservation of natural resources and ecosystems, covering an area of 22.1 million hectares with an additional 5.3 million hectares of marine conservation areas [4]. The area is expected to maintain Indonesia's biodiversity, known as the megadiversity country in the world [56]. Management schemes are also continuously being developed. There are six conservation forest areas: national parks, grand forest parks, nature tourism parks, nature reserves, wildlife sanctuaries, and hunting parks (Law Number 5 of 1990; Government Regulation Number 13 of 1994). In the context of integrating the conservation of natural resources and ecosystems into sustainable development, biosphere reserves are being developed in Indonesia with national parks as core zones [57]. To protect this biodiversity and natural resources, the government implements a conservation program aimed at protecting and restoring plant and animal habitats, preventing species extinction, improving ecosystems, and protecting biodiversity. This has become the focus of the government and all stakeholders in forest management, in situ and ex situ.

Conservation experts offered various recommendations, one of which was from the Director General of Natural Resources and Ecosystem Conservation (2017–2022) in his book "Ten (new) ways to manage conservation areas in Indonesia: developing learning organizations." The ten ways are: (1) Community as a Subject; (2) Respecting Human Rights; (3) Collaboration Across Echelon I of the Ministry of Environment and Forestry; (4) Cooperation Across Ministries; (5) Respecting Cultural and Customary Values;

(6) Multilevel Leadership; (7) Scientific-Based Decision Support System; (8) Resort (Field) Based Management; (9) Rewards and Mentorship; and (10) Learning Organization [58]. Several approaches are applied to manage conservation areas in Indonesia, such as the Integrated Conservation and Development Project (ICDP), the Integrated Protected Area System (IPAS), the concept of bioregional management of conservation areas, the concept of participatory management or community-based management, the concept of forest management with communities, partnership management, and integrated national park management. Still, this approach has not provided optimal results [59].

*3.4. Watershed Management and Forest Rehabilitation*

Watershed management (WM) is closely related to achieving sustainable forest resource management. Water is considered the integrator of all components in the watershed, including the forest ecosystem. Consequently, deforestation will degrade watershed performance significantly and disrupt the quality and availability of water supplies.

The process of social dynamics significantly influences watershed management in the community, which has a consequential impact on changes in the biophysical condition of the watershed [60]. It is crucial to integrate traditional knowledge and local wisdom from planning to monitoring and evaluation to ensure long-term watershed management's sustainability [61].

Watershed management focuses on improving welfare with a broader scope than land resource management or participatory and integrated watershed management [62]. On the other hand, the watershed is the most important planning unit that integrates water and land resource management due to the close relationship between soil, vegetation, including forests, and the water cycle. Globally, most drinking water sources come from forested areas [63]. Hence, appropriate forest cover in a watershed plays a vital role in maintaining ecological stability related to the cycle of nutrients, minerals, energy, and the balance of nitrogen, oxygen, and carbon dioxide gases [64]. Under certain conditions, forests also control water-related disasters such as floods, droughts, and landslides, making forest resource management a vital part of WMP [63]. Several regulations cover watershed management activities, ranging from the state constitution to local government regulations. WM activities are regulated in Government Regulation No. 37/2012, and the main responsibility lies with the forestry sector institution, namely the Ministry of Environment and Forestry. In the regulation, the stages of WM include planning, implementing, monitoring, evaluating, directing, and controlling [65].

3.4.1. Watershed Management Planning

The Minister of Forestry Regulation No. P.60/Menhut-II/2013 regulates Watershed Management Planning (WMP). At the planning stage, the government observes the importance of integrating interests between sectors and administrative areas. This process expects the involvement of all stakeholders and is coordinated by the National/Regional Development Planning Agency. The planning stage generally covers the scooping stage, characterization, including identifying and prioritizing problems, determining management objectives, and defining technical management strategies and practices in all aspects (hydrology, land, socio-economic) [66]. At the scoping stage, forming partnerships between various stakeholders is essential [67]. Several improvements need to be focused on, such as encouraging stakeholders to participate in WMP, including the active role of the local community through appropriate representation. They are expected to assist in identifying potential, capacity, and existing problems to determine watershed management goals.

Documents of the WMP are ratified by the minister or regional leader with the hope that they can be referred to as a part of the Regional Spatial Master Plan (Rencana Tata Ruang Wilayah/*RTRW*) [68]. It has been stated in Government Regulation 37/2012 and Law 17/2019 that the RTRW should incorporate watershed management (Junita and Buchori, 2016). Therefore, it is necessary to internalize the watershed management plan into the RTRW, referring to the implementation of certain programs and activities as planning input

via a series of directed institutional coordination among all stakeholders [69]. To internalize the WM plan into the RTRW, synchronization and connectivity of legislation, policies, and activities between institutions are required. Stronger or higher regulations are needed so they can be implemented at the district and provincial levels. Hierarchical confusion between certain regulations and the asynchrony of authority are also obstacles that must be overcome [70]. Implementers, planners, and regional development decision-makers also play a crucial role. Planners can support these changes by emphasizing watershed concerns more strongly in spatial planning and integrating economic and environmental goals through strategic watershed management [71].

3.4.2. Implementation of Watershed Management

A watershed management information system in each province supports the implementation of watershed management. This system was built and managed by the minister, who organizes government affairs in the field of watershed management by involving related agencies [65]. Forest rehabilitation and reclamation as a form of WM implementation have been regulated in Government Regulation No. 26/2020. Forest and Land Rehabilitation (FLR) aims to restore, maintain, and improve the functions of forests and land to increase their carrying capacity, productivity, and role in maintaining life support systems. Meanwhile, forest reclamation aims to repair or restore degraded forest areas so that they function optimally according to their designation. This regulation states that FRR is carried out through political, social, economic, and ecosystem aspects. FRR is prioritized on degraded land based on specific biophysical conditions as a form of land rehabilitation.

According to Minister of Environment and Forestry Regulation No. 23/2021, forest rehabilitation is implemented through reforestation activities and soil conservation techniques. Reforestation is carried out in two patterns: the intensive pattern, where there are no agricultural activities in forest areas, and the agroforestry pattern, which is carried out in forest areas with community activities. The implementation of reforestation activities consists of several stages, which are: planning, preparation, provision of seeds, planting, and maintenance. Forest rehabilitation is carried out in the following areas: conservation forest (to restore ecosystems, habitat development, and increase biodiversity); protected forest (to restore watershed hydrological functions and increase production of non-timber forest products and environmental services); and production forest (to increase production area productivity).

The MoF has issued Circular Letter Number SE.6/PDASHL/SET/DAS.1/9/2019 concerning the implementation of forest and land rehabilitation through the natural resources conservation business model (*Usaha Pelestarian Sumberdaya Alam/UPSA*). The UPSA model is a demonstration unit of vegetative and mechanical conservation in restoring the carrying capacity of the watershed, conserving natural resources, and improving community welfare. Based on the objectives of the UPSA, the selection of plants is prioritized based on woody plant species and/or non-timber forest product (NTFP) plant species, combined with seasonal plant species. The plant selection processes involve the community as a form of participatory implementation to accommodate community needs (Kementerian Lingkungan Hidup dan Kehutanan, 2018). One of the choices is Multi-Purpose Tree Species (MPTS), a species that can meet the needs ecologically, economically, and socially of the RHL program [44].

The UPSA model was applied in several watersheds in 1985 by the Directorate General of Forest Protection and Nature Conservation, MoEF, together with the Indonesian National Coordinating Agency for Survey and Mapping (Bakosurtanal) [72]. Referring to Edward [73], in 2011, there were 45 UPSA conservation farmer groups in North Bengkulu Regency. However, their implementation did not run optimally due to several problems, such as business capital, farmers' behavior prioritizing the species for their basic needs, crop failure, and price fluctuations, causing farmers to suffer losses. Dwiprabowo and Ginoga [74] found that in Solo Watershed, Central Java Province, the productivity of UPSA's

land is relatively high and can increase income for the community, but the sustainability of the UPSA model is still relatively low.

Currently, the UPSA model is being re-implemented in several watersheds in Indonesia, such as on degraded lands in the Limboto Watershed, Gorontalo Province, starting in 2019 through an agroforestry system by understanding land capability and suitability [75]. The UPSA was also demonstrated as a priority watershed in the Indragiri Rokan Watershed, Riau Province, and the Brantas Watershed, East Java Province. The UPSA demonstration unit in the Indragiri Rokan Watershed, as the only demonstration unit in Riau Province, is developed through a collaboration scheme between the community and the Center for Watershed Management and Forest Protection (BPDASHL) of Indragiri Rokan. The UPSA demonstration unit in the Brantas Watershed in Oro-oro Ombo Village is a pilot unit for an integrated farming system [76].

In 2020, the Ministry of Environment and Forestry (MoEF) carried out environmental and forestry development in the field of watershed control and protection forests, which is carried out using a Holistic, Integrative, Thematic, and Spatial (HITS) approach. In 2020, MoEF carried out land rehabilitation by increasing the vegetation cover by 113 million ha. However, the amount of vegetation cover is not significant compared with the total area of the watershed that must be restored, which is 106.9 million ha [77] and compared with the current deforestation rate, which is 115,460 ha/year [78]. This failure is inseparable from the failure of IWM (Integrated Watersheds Management) implementation, as stated by Basuki, Nugroho, Indrajaya, Pramono, Nugroho, Supangat, Indrawati, Savitri, Wahyuningrum, Purwanto, Cahyono, Putra, Adi, Nugroho, Auliyani, Wuryanta, Riyanto, Harjadi, Yudilastyantoro, Hanindityasari, Nada, and Simarmata [68]. They recommend that the IWM approach be improved holistically, considering all iterative steps of watershed management, including planning, implementation, monitoring, and evaluation. The revised IWM strategy now takes into account the relationship between repairing damaged watersheds and reducing the effects of climate change. Coordination, participation, and collaboration should be prioritized in all management processes in order to improve the IWM for climate change mitigation and adaptation.

### 3.4.3. Monitoring and Evaluation (MONEV) of Watershed Management

MONEV of watershed management is stipulated in Government Regulation No. 37/2012 on Watershed Management [65] and its derivative, i.e., Ministry of Forestry (MoF) Regulation No. P.61/Menhut-II/2014 on Watershed Management Monitoring and Evaluation [79]. Based on the regulation, MONEV of watershed performance is conducted to determine whether the objectives of watershed management have been achieved and to assess the carrying capacity of the watersheds.

Based on MoF Regulation No. P.61/Menhut-II/2014, there are five criteria for evaluating watershed performance: (1) land, (2) hydrology, (3) social economics, (4) building investment value, and (5) regional space utilization. Although it is not explicitly mentioned in the regulations, three of the five criteria are related to forest management, i.e., (1) land, (2) hydrology, and (3) regional space utilization. The components of land criteria, including the percentage of degraded lands, the percentage of land covers, and the erosion index, or the value of land management, are highly tied to forest management. In MoF Regulation No. P.61/Menhut-II/2014, a watershed is in good condition if the percentage of degraded lands is between 5 and 10%, the permanent vegetation is between 60 and 80%, and the erosion index is between 0.5 and 1.0.

The hydrology criteria are related to the land criteria; thus, they link to forest management. The quantity, quality, and spatiotemporal distribution of water yield in watersheds depend on upstream forest management. Proper management of forests in the upper watershed influences water resources at the headwaters and water yield downstream [80,81]. The percentage of protected areas as one of the regional space utilization criteria reflects the importance of sustainable forest management. In this regard, forests regulate water flow and control soil erosion and sediment concentration [82,83].

The results of MONEV using those criteria reflect forest management in the watershed. If the results are good, proper sustainable forest management has been applied. Discontinuous water flow, which results in flooding in the rainy season and drought in the dry season, reflects improper forest management. Currently, MONEV of watershed performance is separated from the implementation of forest management, while these two aspects can be integrated to achieve sustainable forest management. According to Government Regulation No. 23/2021 on Forestry Management, MONEV is conducted for forest inventory control and forest area utilization approval [65]. Regarding forest inventory control, monitoring activity is undertaken to gather data and information on the implementation of forest inventory, while evaluation activity is conducted to assess the implementation of periodic forest inventory based on inventory level. In the future, the MONEV of watershed performance should be linked to forest management to achieve the goal of sustainable forest management. In addition, integrated MONEV of a watershed and forest management should pay more attention to transboundary watersheds or river basins, which are more complex due to the involvement of two or more countries with different characteristics [68].

Traditional knowledge is an aspect that affects the sustainability of watershed management. MONEV of watershed management should consider the integration of local knowledge related to disaster mitigation and forest management and their ecosystems, such as in the community at Mount Merapi [84], community-based land rehabilitation initiatives in the watershed area in Temanggung [85], and local knowledge-based disaster risk mitigation in Gunungkidul [86]. Likewise, the community in the Mutis Mountain Forest catchment area on Timor Island maintains the concept of the "triangle of life" in forest management, namely *Mansian Muitnasi Nabua*, which means that humans, forests, and livestock are an inseparable part of life [87]. In addition, they have local wisdom related to the protection of land and water sources in dry areas of Timor Island [88,89].

The evaluation of watershed management should encompass all components that affect the performance of watershed management, not just the program. Mapping and evaluation of several local wisdoms in natural resource management [90] will assist disaster mitigation through a comprehensive sociocultural and ethnoecological approach from upstream to downstream [91], including the ecosystem services it produces [92]. Thus, community participation and its institutions in MONEV can reduce conflicts of interest in watershed utilization as living spaces, so watershed management can be carried out effectively and efficiently [93]. Relevant to the evaluation of watershed management, several aspects determine human behavior towards the environment, including fundamental factors that involve community perspectives, norms, beliefs, and habits in interacting with the environment [94]. In addition, education, work, culture, and social class factors also contribute, including media factors that encourage ecological literacy and make people aware of the importance of forests and their ecosystem protections [94].

### 3.4.4. Forest Rehabilitation

Current State Regulations on Forest Rehabilitation and Reforestation

More than forty-five years ago, Indonesia started its nationwide program of rehabilitation and reforestation as enforced by Presidential Instruction No. 8/1976. The target of this nationwide program was to conduct land rehabilitation and reforestation on a watershed boundary basis. Various funds and national campaigns were delivered on forest and land rehabilitation activities, i.e., large and massive tree planting components. During this period, 150 official rehabilitation projects have been implemented in 400 locations in Indonesia [16]. The area of degraded and very degraded land throughout Indonesia from 2018 to 2020 is 14,006,450 ha. The realization of forest, land, and mangrove rehabilitation in 2020 is 112,419.41 ha [95]. However, many reports stated that the budget for the programs was spent on ineffective programs with low ecological and social impacts [16].

Government Regulation (PP. No. 26/2020) regarding Forest Rehabilitation and Reclamation (FRR) was launched to promote more serious rehabilitation activity, which is an

essential turning point in implementing a different approach to forest rehabilitation. During this period, the execution of conservation farming on sloping land using soil and water conservation methods, which combine vegetative and civil engineering techniques, became the most effective and commonly used system, especially on Java Island. Since the enactment of the TGHK (Forest Use Consensus) in 1984, rehabilitation activities in protected forests and conservation forests have been advanced by improving ecological functions in protected forest areas. In conservation forest areas, conservation aims to expand biodiversity. However, this endeavor is still unsuccessful because it collides with competition for other uses and increases the number of people who need land [16]. After the 1990s, degraded land rehabilitation was aimed at improving the community's prosperity and increasing wood production for raw materials in the forestry industry. In addition, the government has begun to develop a large-scale Industrial Plantation Forest (HTI) program and rehabilitate degraded land. Still, the rate of forest degradation in Indonesia continues, particularly during the transition from a centralized to a decentralized government system [16].

In the reform era after 2000, based on Decrees No. 136/KPTS/DIR/2001 and No. 001/KPTS/DIR/2002, the rehabilitation program through the CBFM program began to be implemented with changes that occurred from the management of a state-based system focusing on timber production into community-based forest resource management. Forestry Regulation of Indonesia No. 105/2018 described the conservation program covering plantation conservation, soil and water conservation practices, and community empowerment [70].

In the implementation of forest and land rehabilitation so far, watersheds have always been the basis used as a management unit because they give more holistic insight, can be used to evaluate the relationship between biophysical factors and the intensity of social, economic, and cultural activities from upstream to downstream, and are a rapid and easy way to evaluate impacts on the environment [4]. The government keeps updating the regulations up to 2021 on Forest Rehabilitation and Reclamation, Technical Guidelines for Forest and Land Rehabilitation, General Patterns, Criteria, and Standards for Forest Rehabilitation and Reclamation, and Implementation of Rehabilitation of Forest and Land for dryland forest, peat forest, and mangrove forest in forest areas and non-forest areas [6,96]. Nowadays, because population growth affects land conflict in many areas of the country, it is crucial to note that forest and land rehabilitation policy and regulation should be directed to provide good ecological, economic, and social benefits, as well as sustainable agricultural production for better livelihood in the forest community.

The Role and Position of Various Stakeholders Involved in the Rehabilitation and Reforestation Program

The government of Indonesia is the main actor driving the nationwide forest rehabilitation and reforestation program through the Ministry of Environment and Forestry. Not only mineral and dry soil, but currently, the program includes rehabilitation of peat and mangrove areas. These programs are managed under the Directorate General (DG) of Watershed Management and Protection Forest (Dirjen PDASHL) and operationalized at the field site by technical implementation units. Basically, the central government, together with provincial and district/city governments, has been playing the major role in implementing the rehabilitation programs, while the implementation of forest and land rehabilitation in areas with permits/rights is carried out by the holders of permits/rights [16]. At the implementation scale on the site, various non-governmental stakeholders have also been engaging, including the village/local community [97], non-profit organizations [98], formal religious school groups [99], and the coastal community [100].

Important Consideration of Species Selection in Mainstreaming Rehabilitation

Tree species selection and combination are crucial steps in rehabilitation programs for integrating ecology and socio-economic functions [101,102]. Ecologically, native tree species are better than fast-growing exotic species. However, fast-growing exotic species

can speed up the recovery of forest functions [101,103]. Furthermore, rehabilitation using native tree species has also been recognized for its success at the field scale [104,105]. Mixed planting in rehabilitation could have more advantages in terms of biodiversity, economy, and forest health [106] due to higher tree composition supporting a more stable ecosystem with higher resistance to disaster [107,108].

The origin of genetic materials was hardly taken as an important consideration in this nationwide rehabilitation program, yet the genetic composition of the reproductive materials will significantly affect rehabilitation and reforestation success [109]. In the long term, adaptive genetic diversity will promote successful reproduction, reduce the risk of inbreeding and genetic impoverishment, which can result from genetic drift, and increase the population's ability to adapt to future site conditions [110,111]. When seed from a small population is collected, mating among relatives increases, and the consequence will be elevated inbreeding rates. Closely related individuals will have reduced vigor and be vulnerable to pests and diseases, especially in today's rapidly changing, vulnerable world. Many forestry experts have realized the importance of this genetic basis's quality [112–114]. However, regarding the field scale, program planners and implementers, in most cases, did not correctly address the genetic aspect of tree planting. The effect of this misleading conception of not using the proper genetic composition when producing seedlings in a nursery and planting them in the field will not show an immediate effect, but it will accumulate over time as a serious latent problem.

As this concern keeps emerging, MoEF has been triggering various stakeholders to develop seed sources. These seed sources are managed to maintain and improve genetic variation and to produce frequent, abundant, and easily collected, high-quality seeds. The government has also issued regulation P. SK.14/PDASHL/SET/DAS.2/4/2020 concerning the establishment of village planting stock gardens (KBD). It provides for the needs of key native tree species in the area and makes an effort to increase the involvement and livelihood of the surrounding community.

A Way Forward for Forest and Land Rehabilitation in Indonesia

A way forward strategy should be strictly implemented to cope with the inferior soil characteristics and improve the growth of forest rehabilitation plantations, including the application of sufficient organic and inorganic fertilizers [115–117], the application of effective mycorrhizal fungi and bacteria [118], and the introduction of a local fast-growing legume cover crop [101].

Land rehabilitation does not occur in isolation from people. The pattern of the rehabilitation approach requires a change from one that was initially like teaching to a pattern of mutual learning among the involved stakeholders. A partnership pattern must be built between the community and other stakeholders. When communities generate more benefits by protecting the landscape rather than exploiting it, they are more likely to engage in restoration and rehabilitation activities. Those local communities should be considered for more access to restoration activities to promote the success of conservation efforts. When rehabilitation activities are integrated with community livelihoods, site-specific best rehabilitation practices can be developed [119,120].

### 3.5. Social Forestry and Environmental Partnership

Policy changes in forest management have taken place in Indonesia, aiming to expand social forestry (SF) allocations from less than 1% (1.1 million hectares) to more than 10% (12.74 million hectares) of the forest area [121,122]. SF is a sustainable forest management system that is implemented in the state forest area, private forests/customary forests by the local community or customary indigenous community as the main actors. It aims to improve well-being, environmental balance, and socio-cultural dynamics in the form of community forestry, community plantation forests, village forests, forestry partnerships, and customary forests [123]. Many studies highlight the three main principles of social forestry: efforts to grant rights to local communities, support livelihoods, and achieve

conservation outcomes [124,125] for 48 million people in 41,000 villages living within or bordering state forests [126].

SF is also set as one of the National Strategic Programs in the context of Economic Justice to reduce the distribution gap in forest resource use and to decrease unemployment and poverty [22,127]. SF is a compromise approach between the state and the community interests that interact with the boundaries of the forest area, where access to land rights is granted without providing full management authority [124,128,129]. The SF program is implemented with the agroforestry system, which has a potentially important role in increasing farmers' income and sustainable landscape management [22]. The following sections will discuss four important parts of the SF and Environmental Partnership implementation in Indonesia.

### 3.5.1. Historical Background

The SF program in Indonesia adopted and mainstreamed the CBFM concept [130] to address several forest management problems, especially those related to the involvement of the forest-surrounded community, which highly depends on the forest resources for their livelihood. SF is a new paradigm of forest management that considers the social aspect [131], shifting the management approach from timber-based to forest-resource-based management and from state-based to community-based [16]. SF originated from initiatives on the application of institutional science and politics, which aimed to increase the role of local communities in forest resource management [127,128]. SF is held in order to answer tenurial problems and provide justice for local communities and indigenous peoples located around and within forest areas [132].

SF was initiated at the 8th World Forestry Congress in 1978 in Jakarta with the theme "Forests for People," where community rights to forest resources have received greater attention among policymakers, bureaucrats, scholars, and forestry practitioners at the national level [133]. However, community involvement in forest management in Indonesia has existed since the 1960s, especially in the forests managed by Perhutani on Java Island through an intercropping system. Since 1972, various approaches to community involvement in forest management have been developed, such as the prosperity approach, Forest Village Community Development (1982), Social Forestry (1984), Integrated Forest Village Community Development (1994), and Collaborative Forest Management (2001) [134].

The earliest SF scheme introduced by the Indonesian government was Community Forestry (CF) in 1995 through Forestry Minister Decree (FMD) No. 622 of 1995 [135]. Subsequently, through Government Regulation Number 6 of 2007 [136], they began to introduce new social forestry schemes, namely Community Plantation Forests (CFP) in 2007 with Forestry Minister Regulation (FMR) No.P. 23/2007 [137], Village Forest (VF) in 2008 (FMR No. P.49/2008) [138], Forestry Partnership (FP) in 2013 (FMR No. P.39/2013) [139], and Private Forest (PF) in 2015 through Minister of Environment and Forestry Regulation (MEFR) No. P.32/2015 [140]. In order to accelerate the implementation of SF, since 2016, all arrangements for community involvement schemes in forest management (CF, CPF, VF, FP, and PF) have been combined and simplified into one in MEFR Number P.83/2016 concerning Social Forestry [141]. In production forests, all SF schemes can be applied except CtF. In protected forests, social forestry can be carried out with schemes of CF, VF, and FP, and in conservation forests, SF can only be carried out with an FP scheme, while the customary forest scheme is implemented in customary areas. The regulation of SF in Indonesia is very dynamic. Some new regulations were released to improve the previous ones. The latest regulations on SF are Government Regulation No. 23 of 2021 on Forestry Management and Minister of Environment and Forestry Regulation Number 9 of 2021 concerning Social Forestry Management. The achievement of the allocation area of 12.7 million hectares by May 2023 was 5.384 million hectares, involving 1.202 million households and 8150 license units (data obtained from the Directorate General of Social Forestry and Environmental Partnership, MoEF).

3.5.2. Area Preparation for Social Forestry

Given its multidimensional objectives, granting SF approvals must be safe and well-targeted both in terms of community and area for SF development. The targeted community must be those who live in and/or around the forest area, as evidenced by the identity card, and have a social community in the form of a history of cultivating forest areas [123,141]. Meanwhile, in the preparation of the area for SF, it should ensure there is no overlapping status to minimize incidents of tenure conflicts.

Based on Government Regulation Number 23 of 2021, the area for SF is determined by the Minister of Environment and Forestry (MoEF) in the form of an Indicative Map of Social Forestry Area. Granting SF approval is based on PIAPS performed by harmonizing maps owned by MoEF with maps owned by NGOs and other sources and through consultation with the provincial and local governments and related stakeholders. The stipulation of PIAPS is revised every 6 months by considering the dynamics of spatial planning and changes in designation and forest function. Currently, PIAPS revision has been carried out seven times, and in Revision VII, the PIAPS area is set at ±14,677,386 ha [142].

Although PIAPS has been a reference in determining a location for SF, there are still many challenges in its implementation, including frequent tenure conflict incidents and a lack of consideration for biodiversity support [143–145].

The main cause of this mismatch between the indicative map prepared in PIAPS and the actual condition is mentioned by Fisher et al. [134] as a gap between "policy imagination" and implementation experience. Several studies [30,122,146] indicate that incompatibility between maps used as a reference and real situations has become an inhibiting factor for accelerating forest tenure reform in Indonesia. Accountability mechanisms, especially spatial ones (boundaries and mapping), continue to become the main stumbling block in their implementation. Poor data management and integration caused PIAPS not to describe actual land status [147]. Nurfatriani and Alviya [148] reported that the biggest technical problem (50%) is the difference between PIAPS and the location that is proposed by the local government, leading to violations of rights and rules in the field and generating conflicts [128]. In natural resource management, clarity and certainty of tenure status will increase community responsibility for managing natural resources. Tenure security is not just defined as property rights as in the bundle of rights theory but rather as access to benefit from a resource, which refers to the bundle of power theory. Tenure comfort will ensure the potential benefits that will be received as well as the risks that must be faced from the program [21,149].

To avoid overlapping claims on land, each program's implementation must begin with the processes of identification, verification, and area classification to detect potential overlapping status. In addition, a harmonious process also needs to be carried out between the SF area reserved through PIAPS and the community proposal map (if any). 'Free and Prior Informed Consent' (FPIC) needs to be carried out before determining the area for SF to ensure that the rights of the people who have managed the forest area before the determination of PIAPS are not neglected. Hence, any potential conflict can be negotiated and anticipated [21,150]. FPIC is an instrument in international law to protect the rights of communities that are potentially affected by program development. FPIC allows the community to make free choices without coercion before a certain activity is carried out, understand all information about the activity and possible impacts, and express consent. Subsequently, Mohammed et al. [151] said that apart from certainty and security of forest resources, there are other aspects that should be considered as well in determining social forestry areas, i.e., (i) production continuity; (ii) conservation of flora and fauna; (iii) diversification of economic benefits; and (iv) good governance in the formation of institutions.

Finally, identifying key issues and discussions related to jurisdictional boundaries, complemented by field data collection initiatives through participatory mapping, should become the major concern of related stakeholders, leading to the consolidation of the official district and state forest boundary maps. Enriching the PIAPS with other relevant

information such as land status, land slope, potential conflicts, and strengthening the role of the Working Group for the acceleration of SF and conflict management are prerequisites for preparing SF areas. Following this process of implementing activities at the local level and considering the real conditions on the ground is the first step to realizing tenure security and success in SF area preparation.

### 3.5.3. Business Development of Social Forestry

Communities holding SF licenses and members of forest farmer groups (KTH) were registered in a social forestry business group (KUPS) that operates the management of businesses, institutions, and areas to improve community welfare and forest sustainability.

According to data obtained from the Directorate General of Social Forestry and Environmental Partnership, MoEF, up to October 2022, there were 9926 units of KUPS, which were classified into four levels: blue/early stage (50.12%), silver/moderate stage (42.84%), gold/advanced stage (6.49%), and platinum/independent stage (0.55%). Most of the KUPS are still in the early and moderate stages, and only a small part is included in the advanced and independent categories. Therefore, further development is needed to promote the early and moderate KUPS stages into advanced and platinum categories to create more productive KUPS and provide a significant source of income for the KUPS members.

The commodities developed by the Social Forestry Business Group include timber and non-timber forest products and environmental services in accordance with local potential and forest function managed by the holder of social forestry approvals. The dominant commodities cultivated by KUPS are non-timber forest products (NTFPs) at 79.67%, environmental services at 11.75%, and timber forest products (TFP) at 8.58%. The KUPS contributes significantly to the national economy, where the economic transaction value of the KUPS has increased every year in 2018 (IDR 2.075 billion), 2019 (IDR 5.034 billion), 2020 (IDR 17.563 billion), 2021 (IDR 26.745 billion), and 2022 (IDR 159.949 billion) (https://gokups.menlhk.go.id/public/bagan/produksi, accessed on 29 March 2023).

Managing a business as a series of activities must focus on improving economic outcomes that support the community's prosperity enhancement. There are various challenges and opportunities in SF business development, among which are:

1. Lack of forestry extension workers and social forestry assistance. Tajuddin [152] revealed that the weak institutional and low community capacity of SF permit holders caused the low level of activity. Galudra [153] stated that the obstacles to SF business development are insufficient extension workers and social forestry assistance and their roles and capacities.
2. The synergy of social forestry business groups with village-owned enterprises (BUMDES) or other business groups based on village potential is required. Expanding networks and partnerships will facilitate product marketing and increase the added value of the products produced. Budi et al. [154] revealed that access and network capacity levels would determine and influence the success of managing SF businesses.
3. The development of KUPS must be supported by human capital and social capital. Basri et al. [155] and Aritenang [156] confirmed that social capital and human capital with capacity building and community participation are needed for business development based on local groups, such as village-owned enterprises.

### 3.5.4. Environmental Partnership

The environmental partnership policy in Indonesia aims to encourage increased stakeholder roles in environmental and forestry management [123]. This policy is in line with the performance target of the Directorate of Environmental Development, namely increasing the Social Forestry (SF) Group Partnership, the Environmental Partnership, and the number of Social Forestry Assistance Personnel. Some of the strategies to achieve this performance are mapping potential partners, meeting with parties that become partners to discuss

cooperation plans, synchronizing activities between parties who will partner with social forestry groups, and monitoring and evaluating the partnership process [40].

Partnership programs in social forestry have an impact on farmers' income and reduce forest disturbances such as illegal logging and forest fires [157]. This illustrates that cross-sectoral partnership programs are strategic instruments to achieve sustainable natural resource management [158–160] by aligning vision, goals, and appropriate actions [161]. The effectiveness of partnerships plays an important role in achieving sustainable development goals in all dimensions, including environmental issues [162], thus requiring access to resources, technology, and expertise [163], as well as appropriate partners at various levels [164] Environmental partnerships already exist, but SF policies for livelihoods and forest governance generally have not fully met local interests [23]. Integrating central SF and regional policies is one of the strategic options that can support achieving goals.

The community's involvement in forest management will make it easier to monitor forest management because the community itself is involved in the management and receives benefits from the social forestry program. This shows the importance of the work of communities around forest areas [165]. The local community controls a clearly and legally defined area and is supposed to be free from all sorts of immediate state influence on resource utilization [166].

### 3.6. Law Enforcement

Law enforcement in Indonesia is frequently debatable because it is not successfully implemented or can be categorized as a failure. Law enforcement failures are caused by focusing on the target achievement of legal products (producing various laws and regulations) rather than the implementation of the existing regulations [167]. Moreover, the sanction is weak and has not had a deterrent effect on actors [168]. In addition, law enforcement officers focus merely on legal certainty but neglect its benefits and justice [169]. Law enforcement of the environment is getting weaker after the government issued Law No. 11 Year 2020 regarding Job Creation [170]. The law is not maximally implemented when facing the mining companies, and the existing regulation, particularly on biodiversity conservation, has not accommodated the current development on biopiracy [171].

Another reason for weak law enforcement is the absence of synchronization between bureaucracy and rulers, where law enforcement discrimination occurs in many cases. For example, the impediment on law enforcement of forest encroachment in Teso Nilo National Park is disharmony between the Forestry Law (UU No. 41/1999) and other laws. The weaknesses of forest law are articles on crime and its verification, a lack of law facilities and civil servant attorneys, and low community awareness on forest encroachment and sustainability [172].

Therefore, law enforcement in forestry and the environment should be based on the trust stated in the constitution (UUD 1945). To salvage and save forests, it requires special law enforcement [167]. The National Long-term Development Planning (NLDP) year 2005–2025, as stated in Law No. 17 of 2007, has underlined the implementation of law enforcement on forest crime. The condition of forest resources is alarming due to illegal logging, log smuggling, huge land and forest fires, high land demand, wider forest encroachment, and increased legal and illegal mining activities [173].

Irresponsible parties that cause the degradation and deforestation of conservation areas have to be threatened with heavy and strict sanctions. Katimin [172] suggested that policy synchronization can be achieved by synchronizing among law enforcement institutions to trustily enforce the law through an integrated criminal justice system.

Barriers faced by law enforcement on the land utilization of forest areas can be divided into three factors: structural, substantial, and cultural [174]. From the structural side, law enforcers become blind law implementers, focusing on legal certainty and neglecting fairness and benefits. On the substance side of the law, the regulation on forest land use overlapped with its own, among other regulations. From a socio-cultural perspective, law enforcers have not yet understood the cultural values of living in the local community.

This is contrary to the community's perception, pointing out that problem-solving or law enforcement on forest land use is closely related to cultural values in their lives [169].

Another impediment is the weakness of integrated and coordinated law enforcement with limited natural resources. This would affect law enforcers who cannot be implemented according to the law's soul [175]. This weakened government functions, including the sustainability of the environment [176]. Actually, various environmental problems are not strictly affected by the law enforcement applied by the government. The birth of Law No. 32/2009 regarding Environment Protection and Management has not yet answered all problematic environmental law enforcement issues in Indonesia. Although the criminal sanctions in Law No. 32/2009 are more comprehensive than the previous environment law (UU No. 23/1997), law-breaking in the environment still occurred nationally [177]. Therefore, the success of law enforcement should be supported by environment-sound regulations, institutions or organs as law-responsible implementors, and community culture to participate actively and orderly in regulation implementation [170].

The solutions for impediment law enforcement are as follows: First, the out-of-date regulation should be replaced by the new one and supported by related government institutions [167]. Second, increasing awareness of and protecting nature's sustainability is more beneficial than running its tasks, authorities, and protection. Third, the government should control and sanction lawbreakers who violate natural conservation and protection regulations. One of the important things in wildlife protection is community participation (supported by the government and NGOs) as social control over the actions of law enforcement officers. Fourth, publication on wildlife conservation should be intensively conducted both in wildlife domestication (still life) and preservation (dead) as a prestige and home decoration [178]. Fifth, the authorities are willing to apply a criminal law policy approach to forest crimes through a systematic criminal justice system and the empowerment of the criminal justice system. Sixth, the handling of forest crimes should be viewed from various angles. Seventh, the criminal justice agency can cope with forest crimes, especially in law enforcement, using forest encroachment as an ultimate weapon. Lastly, applying a nonlitigation process to solve forest crimes in the field should be pursued [172].

Forest and land fires are crucial environmental problems in Indonesia. The government is trying to control the situation by taking some actions, such as early warning systems monitoring hotspots and more striking law enforcement; however, there are still some gaps, such as inadequate education, inefficient impact assessment methods, and weak coordination between state and community intuition. These conditions have hindered the work of law enforcement agencies [179]. Soedomo and Risdiyanto [180] criticized the law enforcement of forest and land fires. Nurhidayah and Djalante [181] revealed several findings from the study: (1) The institutional and legal framework for managing land/peatland fires is not connected with that for managing forest fires; (2) A lack of law enforcement impedes prevention and mitigation efforts; and (3) The current institutional and legal structures are still primarily concerned with emergency response. Moreover, progress is slower at lower levels of governance, and community livelihood has failed to be integrated into the process. Presidential Instruction No. 11 of 2015 concerning Increasing Forest and Land Fire Control is implemented by enforcing the law involving the military and giving strict sanction to plantation companies and industrial forest plantations that carry out burning in land preparation. This law enforcement can reduce the number of forest and land fires.

### 3.7. Climate Change Control

3.7.1. Indonesia's Commitment to Global Climate Change Control

Indonesia has ratified the Paris Agreement to show its commitment to reducing greenhouse gas (GHG) emissions. The framework under this agreement enables each country to set its own emission reduction target to be compared globally [182]. Law No. 16 of 2016 concerning the Paris Agreement's ratification to the United Nations Framework Convention on Climate Change was further issued as a legal basis for its implementation at the national level. According to this

law, the expected benefits of ratifying this agreement include the protection of Indonesia's vulnerable areas, acknowledgment of the nation's commitment to GHG emission reduction, contribution to decision-making related to the Paris Agreement, and access to funding resources, technology transfer, and capacity improvement to implement climate mitigation and adaptation actions.

Nationally Determined Contribution (NDC)

The achievement of the NDC target cannot be separated from emission reductions in the forestry and other land use (FOLU) sector. According to a recent national GHG inventory, this sector contributes to 50% of the country's total emissions, accounting for 924.8 Mt CO2e [183]. Therefore, the FOLU sector is targeted to have the largest contribution to the NDC emission reduction target compared with other sectors. In the first NDC, the emission reduction target for the FOLU sector is 497 MtCO2e under an unconditional scenario (17.2% of the total emission reduction target) or 650 MtCO2e with international support (23% of the total emission reduction target) by 2030 compared with the business as usual scenario [184]. In the enhanced NDC, which was submitted in September 2022, the target for the FOLU sector increases to 500 MtCO2e (17.4% of the total emission reduction target) under an unconditional scenario or up to 729 MtCO2e (25.4% of the total emission reduction target) with international support [3].

Long-Term Strategy for Low Carbon and Climate Resilience (LTS-LCCR) 2050

Under a low-carbon scenario compatible with the Paris Agreement (LCCP), the emissions are expected to decrease rapidly to be net zero by 2060 or sooner. This scenario sets a target for the FOLU sector to be a net sink of carbon by 2030. According to this document, emission reduction from the FOLU sector includes avoiding deforestation, peat decomposition, and peat fires, as well as increasing carbon sequestration from secondary forests, reforestation, and afforestation [183].

FOLUNETSINK

The FOLU sector emission level is targeted to be $-140$ MtCO$_2$e by 2030 and will continue to increase to $-304$ MtCO$_2$. To achieve this target, policy transformation that supports future land use management should include forest area preconditioning activities to address tenurial issues, conserving the remaining natural forests, improving the regeneration of degraded forests, improving land use efficiency and optimization of unproductive land, acceleration of carbon sequestration, fiscal policy development to increase participation in low carbon development, law enforcement, and database improvement to support the MRV system in the FOLU sector [185].

3.7.2. Regulation on Climate Change

The implementation of climate mitigation and adaptation in Indonesia has been supported by various policies and regulations. To reduce the deforestation rate, the government issued a presidential instruction in 2011 to temporarily suspend new permits in primary forests and peatlands. This regulation was renewed every two years, and finally, in 2019, the government imposed a permanent moratorium on new licenses in primary forests and peatlands. This regulation protects about 66 million ha of forests from deforestation [186]. It is estimated that avoided deforestation in dryland forests after the issuance of this regulation contributed to 3–4% of the NDC emission reduction target [187]. The government also issued Minister of Environment and Forestry Regulation No. 70/2017 concerning the procedures for REDD+ implementation. In regard to peatlands, there is Government Regulation No. 57/2016 amending Government Regulation No. 71/2014 on the Protection and Management of the Ecosystem of Peatlands. This regulation is used as a basis to develop integrated peatland protection and restoration. The mitigation potential from avoiding peat conversion and restoring degraded peatlands is approximately 878 Mt CO$_2$e [188].

### 3.7.3. Instruments Supporting Monitoring, Reporting, and Verification (MRV) for Mitigation
Forest Monitoring System

A forest monitoring system is required to support the MRV system for climate mitigation and adaptation. The Indonesian government has developed the national forest monitoring system, which was first initiated in 2000. This system used remote-sensing technology to monitor annual land cover changes based on national land cover classification. The extent of land cover changes is used to estimate activity data, such as deforestation and forest degradation, following the definition from the Indonesian government. In addition, the forest monitoring system also includes the National Forest Inventory (NFI), which was established in 1989 to support the forest carbon stock database [40].

Carbon Accounting System

The development of a carbon accounting system is an important component of supporting MRV in mitigating climate change and supporting climate-responsible and sustainable forest management. The system is needed to monitor the impacts of land and forest management activities and assess the effectiveness of the climate mitigation outcomes. The Indonesian National Carbon Accounting System (INCAS) is an example of system development that applies the Tier 3 approach [189]. In particular, INCAS has developed a methodological framework for estimating annual GHG emissions from forests and peatlands [190]. The event-driven process is used to quantify the progressive impact of disturbance events occurring on forests (including peat swamp forests), from which GHG emissions/removals are derived. This allows for GHGs to be estimated based on the net change in forest conditions and the nature of the disturbance event that caused the forest to change. For instance, how much biomass was taken off-site and how much remained on site to decay or burn can be determined. This approach tracks the flow of carbon between the different carbon pools in the landscape and ultimately estimates the net GHG emissions released into the atmosphere [191].

National Registry System (Sistim Registrasi Nasional/SRN)

SRN was developed to provide data and information on activities and resources related to climate mitigation and adaptation. This system gathers information from various entities, from the site level to the national level. This system aims to avoid double counting on climate mitigation achievement by implementing clarity, transparency, and understanding principles. Moreover, SRN also serves as the government's recognition of various stakeholders' roles in climate actions and the carbon registry for the implementation of carbon pricing regulation [3,192].

Safeguards Information Systems for Reduction of Emissions from Degradation and Deforestation (SIS REDD+)

The REDD+ Safeguards Information System (SIS REDD+) was developed to collect and analyze information related to safeguards management in REDD+ activities. This system follows seven principles: legal compliance and consistency with national forest programs; transparency and effectiveness of national forest governance; rights of indigenous and local communities; effectiveness of stakeholder participation; conservation of biodiversity; social and environmental services; reducing the risk of reversals; and reduction of emissions displacement. Information on safeguard implementation is first collected at the site level, followed by subnational and national levels. However, this system needs to be strengthened and periodically evaluated, considering the discrepancy in technical capacities among users at various levels [193,194].

SIGN SMART

SIGN SMART was developed to support the national greenhouse gas inventory, implementing Transparency, Accuracy, Completeness, Comparability, and Consistency (TACCC) principles. In general, SIGN SMART aims to provide a database instrument to measure and monitor GHG emissions, provide data and information for GHG emission

reporting, assess emission reduction target achievement at the national and sub-national levels, and provide high-quality data to be used as a basis for sustainable development plan formulation. The information includes the business-as-usual emission level, reporting format, and emission factor database for GHG inventory [195].

SIpongi

The use of the satellite-based system to monitor hotspots in Indonesia has been developing since 1997 with support from the Japan International Cooperation Agency (JICA). In 2015, MoEF developed *SiPongi* to monitor forest and land fires by combining hotspot data from Landsat, NOAA, Terra/Aqua, SNPP, and field verifications. Later in 2019, additional thermal CCTV was installed at 15 priority sites to improve monitoring quality. This system enables early warning hotspot detection, which leads to a faster response to avoid massive fire events [185]. A better fire monitoring system will result in fewer forest and land fire events, contributing to a large amount of emission reduction.

### 3.7.4. Mitigation Action
### Mitigation Roadmap

The climate mitigation roadmap to achieve the NDC target in the FOLU sector covers the improvement of forest resources management through the Forest Management Unit (FMU) in forest areas, the implementation of sustainable forest management practices in production forests, and accelerating industrial plantation forest and community forest development to meet timber demand and reduce dependency on timber harvested from natural forests. In addition, to achieve climate mitigation targets, the government also plans to improve spatial planning to avoid unexpected forest conversion. Increasing land productivity and optimizing the utilization of unproductive lands also become key factors in minimizing the pressure on natural forests while at the same time fulfilling the need for agricultural expansion. Moreover, the national climate mitigation roadmap includes ecosystem restoration, particularly in peatlands, and adopting low-carbon cultivation technology as strategies to achieve emission reduction targets [196].

### REDD+

REDD+ has been a major scheme in the achievement of emission reduction targets in the FOLU sector. REDD+ programs consist of three phases: readiness, implementation, and result-based payment. REDD+ implementation has significantly contributed to emission reduction. It was reported that emission reductions in the period 2014–2016 accounted for 20.26 Mt CO2e under the GCF framework. Meanwhile, emission reductions from REDD+ implementation under the Indonesia-Norway partnership accounted for 17.28 MtCO2e for 2016–2017. REDD+ initiatives at the subnational level also offer result-based payment potential. Forest Carbon Partnership Facilities (FCPF Carbon Fund), which is implemented in East Kalimantan, has a potential payment of 110 million USD in the 2020–2024 period, while the Bio Carbon Fund Initiative for Sustainable Forest Landscapes (BioCF ISFL) in Jambi can potentially receive a payment of 70 million USD in the 2021–2030 period [4].

### Folu Netsink Operation Plan

To achieve the FOLU net sink target by 2030, MoEF has developed several mitigation actions. Protection strategies include preventing deforestation and forest degradation in natural forests and concession areas. Mitigation actions implemented in forest utilization business permit (PBPH) areas consist of plantation forest establishment and improved sustainable forest management through enhanced natural regeneration and reduced impact logging. Meanwhile, forest carbon stock is enhanced by rehabilitating degraded drylands, mangroves, and peatlands using rotational and non-rotational plants. Peatland restoration through the improvement of hydrology management, revegetation, and revitalization of livelihoods is also planned to be improved to achieve the FOLU net sink target. Finally, the

protection of natural forests within and outside forest areas must be strengthened to avoid natural forest conversion for other purposes [185].

### 3.7.5. Adaptation Action
Adaptation Roadmap

A national climate adaptation roadmap was developed based on climate risk identifications, implementation strategies, performance indicators, funding needs and potential fund sources, and stakeholder contributions. In general, the climate adaptation roadmap can be achieved by strengthening policy instruments on climate adaptation and disaster resilience, integrating development plans and fiscal policy, mainstreaming climate vulnerability and risks to various stakeholders, promoting a comprehensive landscape-based approach, building local capacity based on available best practices, improving knowledge management, improving the participation of relevant stakeholders, and adopting adaptive technology [197].

Climate Village Program (Program Kampung Iklim/ProKlim)

PROKLIM combines climate adaptation and mitigation actions at the site level by actively involving local communities and other relevant stakeholders. This program is expected to enhance food, water, and energy resilience, considering long-term climate risks and potential hydrometeorological disasters. Adaptation components in ProKLIM include drought, flood, and landslide prevention, food security improvement, the anticipation of sea level rise, coastal abrasion, and seawater intrusion, controlling climate-related diseases, and other activities to improve climate adaptation [198]. This program was first launched in 2016, and 3270 villages will have been registered under PROKLIM by 2021 [186]. However, lack of capacity and technical support is still an issue in PROKLIM implementation [199].

### 3.8. Human Resource Development

Improving the quality and competitiveness of human resources is one of the seven national development agendas, and it is crucial for national development in Indonesia. As a key component in an organization, human resources development must pay attention to increasing competitiveness, intensifying collaboration, and emphasizing activity results for sustainable development [200,201]. The determination of policies for the management and development of human resources in forestry management in Indonesia is carried out by the Minister of Environment and Forestry, the Director General, and the Secretary-General as key stakeholders, together with BP2SDM (Human Resources Counseling and Development Agency) as the main stakeholder [202]. BP2SDM, as one of the organizational units of human resources development under the Ministry of Environment and Forestry, is in charge of ensuring the availability of adequate human resources, both in quality and quantity, to be able to face the dynamics and challenges of forestry development. Counseling, education, and training are important human resources development activities to be carried out in forest management in the context of economic development in Indonesia [203]. In its performance report, the Human Resources Development Planning Center said that the performance achievement in 2021 was 107.3%, with a budget realization of 99.98%. These performance achievements are based on three activity performance indicators (IKK): (1) one service map of developing human resources competencies for apparatus, (2) one service map of developing human resources competencies for non-apparatus, and (3) one-thousand environmental and forestry human resources increasing their competence [204]. Although the performance achievement is adequate, several efforts are needed to enhance the performance of human resources in forestry management. According to the research by [202], the performance assessment of forestry human resources, especially related to the licensing of forest utilization, the lease of forest area, and the release of forest area, is subpar. It is necessary to form a team to evaluate the performance of forestry human resources, build commitment to providing high-quality services, and develop one-stop licensing to create transparency and accountability for the public. In 2022, BP2SDM supported three of

the six national priorities: (1) strengthening economic resilience; (2) enhancing the quality and equity of human resources; and (3) building the environment, increasing disaster resilience, and addressing climate change. The National Priority is achieved through several activity units at the Human Resources Development Planning Center, including (1) a map of developing environmental and forestry human resources competencies for both apparatus and non-apparatus, (2) competency tests/certification of human resources, (3) development competency standards, (4) supporting human resources competencies through e-learning programs, (5) management support services, and (6) office services [204].

### 3.9. Research and Development Policy and Implementation

3.9.1. Hundred Years of Forestry R&D

Forestry research and development policy in Indonesia could be discussed according to the periods of government regimes: colonial period (1913–1945), *Old Order* (1946–1965), *New Order* (1966–1998), and Reformation period (1999–until now). It was commenced in 1913 by the Dutch colonial government by forming an institution named the Research Station for Forestry under the Forestry Agency [205]. In 1927, the name of the research station changed to the Institute for Forestry Research. The research focused on forest production, forest planting, forest exploration, forest protection, forest hydrology, and wood technology. The Forest Botany Herbarium was built in 1917, and xylarium materials have been collected since 1915. The Office of the Institute for Forestry Research in Gunung Batu, Bogor, was built and inaugurated in 1931 and was completed with a laboratory, library, and workshop. Other research facilities that were built in the period were experimental gardens/research forests at several locations, such as Cikampek (Purwakarta), Pasir Awi (Bogor), Cigerendeng (Ciamis), and Haurbentes (Bogor) [206]. During Japan's colonial period (1942–1945), the name of *Bosbouw proefstation* shifted to *Ringyoo Sikenzyoo*, where the main task focused on wood utilization techniques. Then, after Indonesia became dependent on 17 August 1945, the name of Ringyoo Sikenzyoo changed and returned to the previous name, namely Forestry Research Institute(FRI) [205,206].

Some important research results in the colonial period were a method of log measurement, especially teak wood, and determination of the wood volume through wood volume tables; the conversion score from the staple meter into m$^3$ and kg; the fruiting period of some tree species; a method in the determination of *bonita* (soil fertilization) class for plantation forests; criteria and methods in the determination of wood strength and durability classes [206].

During the *Old Order* period, following the development program of forestry research concerning the policy on forestry industry development planning, some research facilities were gradually provided, such as buildings (office rooms and laboratories), research tools, and institutional development. The research activities also became wider, including pulp, venire, plywood, wood drying, and wood preservation. In 1956, the institution was enlarged and became the Center for Forestry Research with two research institutions: the Forest Research Institute and the Forest Product Research Institute [206,207]. Some tree species were planted in the arboretum, and the experimental garden/research forest was added, among others in Padekan Malang, Situbondo (1952), Cikole, Bandung (1954), Arcamanik, Bandung (1954), Carita, Pandeglang (1955), and Dramaga, Bogor (1956) [206]. The significant research results during the *Old Order* period were utilized in practices such as a stand volume table of 10 tree species with *bonita* class and thinning degree; estimation of teak tree volume based on measurement of stem circle at breast height; and a data set on wood properties as material in preparing a regulation on Indonesian wood construction, which was used by the Public Work Department. The Forestry Research Center also had a role in the supervision of plywood industry construction [206].

At the beginning of the *New Order* period (1966), the Ministry of Forestry was omitted and became the Directorate General of Forestry under the Ministry of Agriculture. Thus, management of the forestry research center was also moved into the Directorate General of Forestry. In 1983, the Ministry of Forestry was formed again. Forestry research is managed

under the Forestry Research and Development Agency (FORDA) with two research and development centers, namely the Center for Forest Research and Development and the Center for Forest Product Research and Development, and an agency secretariat was also formed to coordinate them. In 1993, the scope of forestry research was expanded. The names of the two research and development centers were changed to Center for Forest and Nature Conservation Research and Development, Center for Forest Product, and Center for Forestry Socio-Economy Research and Development. This was followed by the establishment of the Kupang FRI (1993) and the Institute for Seed Forest Plant Breeding in Yogyakarta (1994) [206].

In order to facilitate forestry research, which has evolved and covered many aspects during the *New Order* period, new office and laboratory buildings as well as various machines were established and provided, among others, tissue culture laboratories, sawmill laboratories provided with sawmill machines of commercial factory scale, saw-doctoring laboratories, wood wall machines, wood preservation installations, etc. Other facilities established during this period were two research stations: Wanariset I Samboja in Kalimantan and Wanariset II Kuok in Sumatra; a demonstration plot for rehabilitation of burn forests (1000 ha), an arboretum, and a germplasm garden for Kalimantan native fruit trees. Research collaborations have been conducted with external institutions, both national and international. The scope of the research in this period was extended to support natural forest concession activities, which began in the 1970s, industrial plantation forest establishment (starting in the 1980s), and also the wood industry.

The significant research results on System Silviculture in Indonesia: Selective Cutting and Industrial Plantation Forest development has been used by private timber companies for natural and plantation forests as guidelines in their forestry business. The building industry and the wood industry have applied research results to wood preservation and drying technologies. Other important research results in this period were identifying important tree species used to inventory forest stand potency and tree volume estimation through a tree volume table used to assess forest stand volume and determine the annual allowable cut. Research results on eaglewood, mycorrhizae, apiculture, sericulture, and tree pest control were also used in practices. The expertise of researchers in forests and forest products has been used in the formulation of various forestry product standards and feasibility studies for some forestry industries [206].

The FORDA transformed into the Forestry and Environment Research, Development, and Innovation Agency (FOERDIA) during the era of President Joko Widodo (2014–now) after merging the Ministry of Forestry and the Ministry of Environment into the Ministry of Environment and Forestry (MoEF). FOERDIA has the task of conducting research, development, and innovation in the field of environment and forestry, including disseminating the results of the research, development, and innovation to both internal and external users of the MoEF. FOERDIA evaluates social forestry programs and provides input for improvements to simplify the flow of social forestry permits and provide appropriate empowerment methods for forest farmer groups based on available resources [208].

Forest fire prevention and management research is one of FOERDIA's priority programs. Prevention efforts are considered more effective than firefighting efforts. The community participation approach was conducted by building fire-alert villages. FOERDIA provides criteria and indicators to measure the level of village alertness in controlling forest and land fires so that an early anticipation program can be carried out [209].

Forest and land rehabilitation efforts are a priority program for each period of government. The success of forest and land rehabilitation is supported by the availability of quality seeds. FOERDIA has succeeded in setting criteria for quality seeds. The tree breeding program developed by FOERDIA has supported the formulation of strategies and the implementation of genetic conservation for endangered forest plants [210]. In addition, FOERDIA produces several superior species with improved genetic traits so as to produce high productivity, including *Acacia mangium*, *Eucalyptus pellita*, *Falcataria mollucana*, *Tectona grandis*, etc.

FOERDIA also plays an active role in various international issues, one of which is climate change and carbon trading. FOERDIA also plays a role in the preparation of allometric models for estimating tree biomass in various types of forest ecosystems in Indonesia, which contain estimated values of biomass and tree volume in natural and plantation forests [211]. Indonesia's REDD+ Measurement, Reporting, and Verification (MRV) Guidelines, which contain guidelines for the implementation of MRV, have been implemented by implementers in the field [212]. The research and development agency also publishes a book on REDD+ and forest governance. Based on Presidential Regulation No. 78 of 2021 concerning the National Research and Innovation Agency (BRIN), all research institutions are combined into one at BRIN. Research programs and resources at FOERDIA will eventually shift to BRIN starting in 2022.

3.9.2. Law 11 2019, and BRIN

Some of the fundamental issues of incompatibility between the research conducted and the needs of policymakers in technical agencies for developing evidence-based policies (EBP) in Indonesia include limited research funds, non-linearity of research and policy needs, and an unbridgeable gap between research languages and practical policy languages.

Based on the World Bank report in 2022 [213], the research budget in Indonesia in 2020 was 0.28% of GDP, far below other Southeast Asian countries such as Malaysia in 2018 at 1.04% of GDP, Vietnam at 0.53% (2019), and Thailand at 1.14% (2019). Research results can answer only a few questions about forest management and the environment. Even then, it is generally incomplete because the research is often discontinuous due to funding reasons.

The other chronic problem is that the research topics produced are not in line with the needs of policymakers due to a lack of communication and coordination between research providers (researchers and scientists) and research users (policymakers). Research results can answer only a few questions about forest management and the environment. Even then, it is generally incomplete because the research is often discontinuous due to funding reasons. The next problem is the gap in terms of language. Research results generally lead to scientific journals that use scientific and technical languages. This scientific language still needs further transformation into practical language for policy-level implementation in the field.

Law No. 11/2019 on the National System of Science and Technology, enacted on 13 August 2019, is considered a milestone for Indonesia's EBP development. One of the important articles in this law related to EBP is the one stipulating that the results of research, development, assessment, and application must be used as a scientific basis in formulating and determining national development policies. It is to confirm that EBP is obligatory, not voluntary.

The National Research and Innovation Agency (Badan Riset dan Inovasi Nasional/BRIN) was formed through the issuance of Presidential Regulation (PP) Number 74/2019 (24 October 2019) concerning BRIN to address issues of the fragmentation of R&D organizations and the accountability and inefficiency of R&D programs as a mandate of Law 11/2019 concerning the National System of Science and Technology. Based on PP 74/2019, BRIN was an institution attached to the Ministry of Research and Technology, but through PP No. 33/2021, then replaced by PP 78/2021, BRIN is designated as an independent institution as the only national research institution. The last two PPs emerged as a follow-up to the emergence of Law 11/2020 concerning Job Creation, an omnibus law regulating regulatory changes in various sectors to improve the investment climate and create legal certainty. The Omnibus Law revised and trimmed 80 laws, including Law 11/2019 concerning the National System of Science and Technology.

The fundamental change from PP 74/2019 to PP 33/2021 is the existence of a Steering Committee on the BRIN organizational structure. From the details of the duties of the Steering Committee and the qualifications of the chairman of the Steering Committee, it appears that the existence of a Steering Committee in the BRIN structure is to ensure that research, development, study, and implementation activities are based on the ideology of

Pancasila as the Foundation of the Republic of Indonesia. In PP 78/2021, it is explicitly stated that national research and innovation must be based on the Pancasila ideology.

Another fundamental change is regarding the function of BRIN. In PP 74/2019, BRIN does not carry out an implementing function but only a function of directing, formulating policies, and synergizing, coordinating, and facilitating research, development, assessment, and implementation activities. Meanwhile, through Presidential Decree 33/2021 as well as Presidential Decree 78/2021, BRIN carries out the function of implementing research, development, study, and implementation activities, in addition to the functions of policy formulation, synergy, coordination, and facilitation.

In the environmental and forestry sectors, after the establishment of BRIN, all research and development activities are conducted by BRIN through several research centers under the Life Sciences and Environmental Research Organization. Starting in mid-2021, the Ministry of Environment and Forestry will not be mandated anymore to conduct forestry research and development activities, which have been part of its duties for about three decades.

## 4. Present and Future Challenges

As a country with the 8th largest forest area in the world [214] and the third-largest in the tropics, forests and Indonesian communities are inseparable. Around 25,800 villages (34.1% of the total villages) are directly adjacent to forest areas [4], so most communities living around or in forest areas and indigenous peoples still depend on forest areas. They utilized various natural resources as a source of food, shelter, energy, a source of protein supply, dominant cultural expression [215], and economic and livelihood opportunities.

### 4.1. Climate Change

Forests and their functions play an important role in climate change mitigation and adaptation [216]. Therefore, mitigating climate change through good forest management requires the participation of stakeholders who carry out different activities with the same goal [217]. A complex adaptive forest management concept is needed whose approaches are based on resilience, functional diversity, assisted migration, and multi-species planting as a flexible and multi-scale way to manage forests for human life [218].

As described by Falk et al. [219], Indonesia and Southeast Asia are regions with tropical rainforest conversion into the fastest agricultural land use in the world, whose impacts can affect local climate and water flow. Only a few countries, including Indonesia, have integrated mitigation and adaptation policies in managing climate change [216], despite it being a global issue requiring international consensus in its handling [220]. However, Indonesia has committed to participating in the problem by supporting the Paris Agreement through Law Number 16 of 2016 concerning the Ratification of the Paris Agreement to the United Nations Framework Convention on Climate Change, promulgated on 25 October 2016 [221].

The government of Indonesia prioritizes forest and peatland restoration activities in the NDC by targeting to restore two million hectares of peat in Indonesia by 2030 [222]. Better peatland management, including a moratorium on granting new conversion permits, is the cornerstone of Indonesia's climate change mitigation commitments [223].

Implementing climate mitigation actions in the forestry sector has faced various challenges. Funding is one of the most common obstacles affecting climate mitigation performance. Even though carbon market regulation has been issued to improve stakeholders' participation in achieving the NDC target, the pricing mechanism for various characteristics of climate actions has yet to be available. The payment system of the result-based payment scheme in REDD+ initiatives remains questioned. Not only funding mechanisms but a fair benefit-sharing operation also become issues that need to be addressed [224,225]. Sufficient long-term funding is required to ensure the sustainability of climate mitigation actions [226]. Meanwhile, the carbon market as an alternative funding source has not been proven to provide accessible and sufficient funding, particularly for site-level stakeholders [227].

Another challenge in climate action is intersectoral coordination. The Forestry and Other Land Use (FOLU) Net Sink operational plan noted that the target locations include forest areas and areas for other purposes (APL) under the management of various stakeholders [185]. Collaboration between local and central governments, the private sector, and local communities is strongly required to synchronize program implementation in the field. Furthermore, the expansion of agriculture and bioenergy plantations for renewable energy may lead to land cover changes, resulting in GHG emissions [228,229]. Therefore, integrated planning in the land sector, which includes forestry, agriculture, and energy sectors, is needed to avoid forest conversion and improve productivity in agriculture and plantation areas.

### 4.2. Water, Food, and Energy Scarcity

Rapid population growth causes increased use of land, water, energy, and natural resources to meet people's socio-economic needs [230]. This poses a compelling challenge for us to ensure that the food, water, and energy supplies provided by the Earth are not short-lived for future generations. Water, food security, and energy are inseparable units (nexus), and each will influence the other [231].

### 4.2.1. Contribution of the Forest Sector to Food Security

Law Number 18 of 2012 defines that food security is a condition of food fulfillment for the state down to individuals, which is reflected in the availability of sufficient (both in quantity and quality), safe, diverse, nutritious, equitable, and affordable food supplies that do not conflict with religion, beliefs, or community culture to be able to live a healthy, active, and productive life in a sustainable manner. The derivative regulation to support national food security in Indonesia is Government Regulation Number 23 of 2021 concerning the Forestry Administration, which states that food security in forest areas is part of the national strategic program in order to support national food security.

The Ministry of Environment and Forestry has kept unproductive forests as food reserves. Based on the land cover in the Forest Utilization Business Permit (PBPH) for Natural Forest and PBPH's Industrial Plantation Forest (One Map KLHK, 2021), there is potential for the development of food reserves covering an area of 3,213,476 ha consisting of open land (309,116 ha), Dryland Agriculture/Shrubs (2,903,269 ha), and Ponds (1042 ha). The location of food security land can be built in the PBPH and management rights areas. In order to guarantee a sustainable food production system and environment, one of the prospective land-based models is agroforestry systems with various forms and modifications following local agro-ecological conditions to obtain the suitability of plant species for the land.

For a successful agroforestry practice, we must address another challenge in food security. A major part of the Indonesian population depends on rice as their staple food, and it was projected there would be 15 million tonnes of potential rice imports in 2045 despite the potential surplus production [232]. One potential source of diversified carbohydrate nutrition is tubers, such as cassava, sweet potato, potato, arrowroot, *gadung*, *kimpul*, *taro*, *gembili*, *ganyong*, and so on [233]. Therefore, promoting food diversification is the key to a sustainable output of agroforestry practices by alternating different kinds of staple foods to avoid the negative impact of climate change on rice production.

### 4.2.2. Contribution of the Forest Sector to Water Regulator

As a basic human need, fulfilling water needs is one of the targets of the Sustainable Development Goals (SDGs), as mandated in Presidential Regulation No. 59 of 2017. Achieving the SDGs is necessary to ensure the availability of adequate water in terms of quality, quantity, and distribution. As an archipelagic country, Indonesia's potential for water resources is very abundant [234,235]. However, as the population grows, less water can be stored. According to data from 1996, there is 2110 mm year$^{-1}$ of available water, or 127,775 m$^3$ s$^{-1}$, equivalent to 4 billion m$^3$ year$^{-1}$ [236]. Data from the Ministry of

Environment and Forestry in 2019 [237] shows that the carrying capacity of water resources on the two main islands in Indonesia, namely Java and Bali-Nusa Tenggara, has been exceeded (Table 1). Therefore, it is necessary to carefully manage the potential of water resources to meet the population's needs.

**Table 1.** The water resource carrying capacity of the main islands in Indonesia.

| Island | Water Availability ($\times 10^6$ m$^3$/yr) | Water Utilization ($\times 10^6$ m$^3$/yr) | Max Population That Can Support ($\times 10^6$) | Status of Carrying Capacity |
|---|---|---|---|---|
| Sumatra | 520,502.9 | 178,703.9 | 650.6 | Not over |
| Java | 118,901.3 | 117,613.3 | 148.6 | Over |
| Bali and NT | 20,691.7 | 23,042 | 11.6 | Over |
| Kalimantan | 633,742.80 | 108,054.4 | 792.2 | Not over |
| Sulawesi | 138,071.9 | 54,005.6 | 172.6 | Not over |
| Maluku | 50,005.5 | 8424.2 | 58.9 | Not over |
| Papua | 606,447.3 | 7513 | 758.6 | Not over |

Source: [237].

Physical characteristics (topography, soil, and land cover), climate conditions (precipitation, temperature), population size, and water use behavior affect water availability [238,239]. The demand for water has been increasing through the years, along with the increase in population [240]. On the other hand, floods, droughts, lack of water storage, watershed damage, pollution of water bodies, and various threats to the existence and sustainability of water resources are just a few of the issues with water resources that frequently make it challenging to meet the demand for water [235]. In Indonesia, there are at least three criteria for areas with increasing water scarcity susceptibility: areas with a high population, areas with low rainfall, and small islands. As an archipelagic country, many small islands in Indonesia have the potential to experience water scarcity triggered by limited clean water availability and the threat of water pollution, as happened on small islands in Sulawesi and East Nusa Tenggara [241,242].

An integrated strategy is required to combat water scarcity and create nexuses with other sectors, such as energy and food [243–245]. Integrated watershed management is also part of the strategy to address the water scarcity issue. Its activities include monitoring pollution and water quality; forest and land rehabilitation; caring for springs and other water sources; and enforcing laws, especially concerning water body pollution [70]. The strategy should be prepared in a participatory manner so that it is more targeted. Several studies show that water resilience problems can be solved by involving the community in forest and water resource management [246–248]. Forests are closely related to water resources [249], so forest management must be appropriately considered.

Forests are essential for producing and regulating the water cycle [250]. Ignoring forest functions will make it difficult to assess, adapt, and reduce the impacts of forest land cover and climate change [250]. Another opinion also suggests that land cover (including forest) is one of the determinants of regional water yields [251]. As a regulator of the water system, the forest serves one of its ecological purposes. In this case, the forest can maintain the time and availability of river water flow, support the microclimate, and protect the downstream area from various disasters such as flooding [252].

Research also concluded that annual water yields would increase as dense forest vegetation declined [253]. Furthermore, Bruijnzeel [254] states that the water yield from a certain amount of rainfall in upstream areas with cloud forests tends to be higher than in mountain forests unaffected by fog and low clouds. Moreover, water runoff from cloud forest areas tends to be more stable during common rainfall conditions and occurs over a long period of time [254]. Another consequence of a land cover change in the cloud

forest area is the occurrence of net rain. No more trees can withstand rain or fog, and clear showers will decline [255].

4.2.3. Contribution of the Forest Sector to Bioenergy

Bioenergy is a type of energy derived from biomass. Biomass can be used to generate electricity, transportation fuel, or heat. Traditional biomass is derived from local solid biofuels (wood, charcoal, agricultural residues, and animal dung) that are burned using basic techniques such as traditional open cookstoves and fireplaces. Over the last decade, two concurrent trends can be observed: traditional biomass uses have been slowly declining (−7% in 2008–2018), while the share of modern renewables has gradually increased from 8.2% in 2008 to 10.7% in 2018. [256].

Given the vast land area that can be used to produce various types of biomass, either as residues (e.g., from agriculture, forestry, or urban waste) or alternative products, Indonesia has a high potential for bioenergy development (e.g., vegetable fuels, palm oil derivative products, or certain types of plantations). The government promotes land rehabilitation through planting activities. Community forest programs, social forestry, PHBM, and other similar programs have a lot of potential. Plantation development on degraded lands can increase the economic value of wood-based energy generation in comparison to other energy sources (even renewable ones). The Sustainable Forest Management Program to increase the area of vegetation cover and restore land conditions in the watershed has the potential to support the availability of bioenergy raw materials. The program's goals are met through rehabilitation inside and outside forest areas.

Energy plantation forests can potentially be developed to produce wood biomass as a renewable energy source [237], which can be converted into bioenergy through various technologies and become part of Indonesia's energy mix [257]. The types of energy plants have high heat value criteria, fast-growing species, fast coppicing, and wider adaptability, such as *kaliandra* and *gamal* [258]. The development of energy plantation forests also has the potential to reduce deforestation and forest degradation [259]. Therefore, through PP No. 23 of 2021 concerning Forestry Implementation, the Minister prioritizes accelerating the inauguration/release of forest areas, one of which is procuring energy securities.

The utilization of bioenergy potential still faces a number of challenges, including distribution, material supply continuity, and economic aspects [260]. The increase in renewable energy capacity is constrained by increased investment and limited mastery of technology, which have an impact on non-optimal potential utilization [261]. The focus on biofuel production in Western Indonesia has resulted in an uneven distribution of biofuels, while there is a lack of infrastructure and assurance for biofuels derived from palm oil [262,263]. The low rate of energy conservation adoption by energy users is one of the issues concerning energy conservation. Other challenges are the high investment required for efficient energy applications, the lack of regulatory support for energy conservation investment, limited incentives and disincentives for implementing energy conservation, and the absence of a cross-sectoral monitoring and evaluation system [264]. As a result, strategic steps can be taken for carbon balance analysis, land allocation, land use, sustainable resource use, technology support, a focus on high-added value, and improved governance.

The development of oil-based biofuels, especially in forest areas, needs to pay attention to aspects of environmental sustainability. Palm oil production in Indonesia is attributed to enhancing 28% of forest loss and 44% of forest fragmentation in concession areas since 2000 [265]. Another analysis shows that the sustainability status of palm-based bioenergy is still low in social and economic aspects and moderately sustainable in environmental aspects [266]. Utilization of waste from the palm oil industry (such as empty fruit bunches, mesocarp fiber, and palm kernel shell) as a source of biomass energy is expected to be one of the efforts to increase the sustainability of palm-based bioenergy [262]. In addition, the use of non-edible parts of plants as a source of bioenergy can also reduce food and energy conflicts [267].

### 4.3. Human and Wildlife Conflicts

Forest exploitation, human growth, and climate change have resulted in reduced wildlife habitat [268,269]. Conversion of forest land for transmigration, rubber and oil palm plantations, fires, illegal logging, encroachment, settlements, and road expansion are some of the direct and indirect factors that cause deforestation to continue [270]. Humans and wildlife have to compete for increasingly scarce land resources [271,272], triggering conflicts between them [273,274].

Currently, human and animal conflicts are increasingly widespread, especially in the habitats of endangered and protected animals such as *banteng* (bulls), elephants, tigers, orangutans, monkeys, and saltwater crocodiles [275–279]. For example, human conflicts with elephants and orangutans in Sumatra always occur annually and in almost every metapopulation [280–282]. Human conflicts with saltwater crocodiles happened in East Kalimantan and East Nusa Tenggara, two provinces with the highest crocodile attack rate in Indonesia [275,279]. Fitria et al. [283] also reported that from 2011 to 2019, there was a disturbance of long-tailed macaques in 15 districts in Central Java, involving thousands of monkeys and causing damage to crops and settlements. Conflicts between humans and wildlife have significant consequences for animal and biodiversity protection, human safety and well-being, and ecosystem integrity.

Kuswanda, Harahap, Alikodra, and Sibarani [277] stated that animals entered the company and community land because these locations were their home range and their foraging area. The conflict resulted in a population decline due to being hunted, poisoned, or killed. On the other hand, humans also suffer economic losses due to their crops being destroyed by animals, injury, and even death [272,284,285]. Conflicts between humans and wildlife have significant consequences for animal and biodiversity protection, human safety and wellbeing, and the integrity of ecosystems [286].

Wildlife rescue and conflict management efforts have been carried out by the Indonesian government, assisted by NGOs, the private sector, and other stakeholders. Specific regulations for resolving conflicts are stated in Forestry Minister Regulation No. P.48/Menhut-II/2008, amended to P.53/Menhut-II/2014. In the future, regulations at the ministry level alone for mitigating human-animal conflicts will be considered insufficient because they must involve other institutions from different ministries. Higher regulations, such as government regulations, are needed as instructions for various institutions across provincial and district governments [284]. Human-wildlife conflict mitigation policies in Indonesia must accommodate various interests in forest management, such as animal welfare, social, economic, cultural, political, and provincial/district spatial planning issues [287,288].

### 4.4. Wildlife Illegal Trading

Illegal wildlife trade still occurs because of the poor condition of the people around the forest, who perceive forest ecosystems as natural resources that can be used to increase their income [289–291]. Whether legal or illegal, the trade in wildlife products is one of the most valuable businesses, locally or globally [292]. In the world, the trade involves hundreds of millions of wild plants and animals; 100 million tonnes of fish, 1.5 million live birds, and 440,000 tonnes of medicinal plants were traded in just one year. On the other side, wildlife trading in Indonesia has been about 7.7 million since 1975, including 2 million *arwana* (*Scleropages* spp.) [290,293].

The rise of illegal wildlife trading was due to the jurisdiction of destination countries that permit trade in these species. In addition, the relevant enforcement agencies find it difficult to distinguish between wild-caught and captive-bred animals [294–296], requiring additional forensic DNA evidence from the body parts of this wildlife [297–299]. Efforts and participation from the government (Ministry of Environment and Forestry and Police), the community, and environmentalists (Pro Fauna, Green Peace, and WWF) are needed together to control the rate of illegal trade in endangered species. The use of broadcasting institutions, whether radio or television, as well as various social media platforms to combat the illegal wildlife trade, must be coordinated and utilize local, national, regional, and

international cooperation through CITES [300,301]. Finally, research and development through breeding and captive breeding of species must be directed at conserving the existence of wildlife [291].

*4.5. Forest Management Conflicts*

4.5.1. Conflict in Managing Forests

The government has attempted to involve various stakeholders in forest management to achieve sustainable forest management and improve the welfare of the communities around the forest, but in practice, there have been various obstacles and conflicts. Judging from its history, conflicts in the forestry sector in Indonesia have existed since the colonial period, when the Dutch monopolized teak forests in Java [12]. Historical records also show conflicts that arose in the early 20th century between forest rangers and the community. Meanwhile, forestry conflicts outside Java (Kalimantan and Sumatra) began to emerge in the early 1990s between logging companies and local communities [302,303]. However, small-scale conflicts emerged when the government started issuing permits for logging in the early 1970s [304]. This conflict was motivated by disputes over access rights to forest resources and economic problems.

Land tenure conflicts in Indonesia are complex, involve many actors, and have occurred for decades [305,306]. These actors are local communities, companies, and the government. Many examples of conflicts have occurred, mostly involving indigenous peoples. Conflicts between the government and indigenous peoples occur because they encroach on the forest and claim to own the land [307]. The conflict occurred in Tamiai village, Kerinci Regency, between indigenous peoples and migrants [308]. Conflicts between an oil palm plantation company and local communities in Rokan Hulu Riau were caused by land use changes by the company [309].

Land tenure is the most significant conflict in the Indonesian forestry sector. Conflicts between forestry and palm oil dominated the conflict. Around 2.5 million ha of oil palm plantation areas are located within forest areas, whereas local communities own 1.7 million ha. In comparison, 0.8 million ha were owned by private companies [310], and the number of conflicts related to oil palm reached 1061 cases in 2020 [311]. Furthermore, their expansion increase is very significant, from only 1.1 million ha in 1990 [310] to 16.4 million ha in 2019 [312], due to the government's target to raise economic growth.

4.5.2. Future Challenges in Managing Sustainable and Minimal Conflict in Indonesia's Forest

Indonesia will face the challenge of achieving sustainable forest management with minimal conflict that can accommodate stakeholders' interests in the future. The crucial thing to achieve this condition is to build awareness and concern for the environment among all stakeholders in forest management [313,314]. It is also necessary to increase interest, participation, and commitment [315–317] for the local community from upstream to downstream [315,316]. We can increase the interest and participation of the community and stakeholders by increasing the benefits of resources [121,315], increasing access to financial resources [121,318], creating business orientation [314,315], involving local people in tourism [319,320], technology, and markets [121].

The challenges that also need to be addressed by the government are clarifying the position of customary forests, recognizing communal rights, and settling the boundaries of forest areas managed by indigenous peoples [121]. The government also needs to assist, increase capacity, and empower indigenous peoples [121,321] and local communities [121,314,318,322]. These activities are necessary to maintain and strengthen the existence of indigenous peoples and community institutions [318] in managing forest areas based on customs and local wisdom.

Forest management with minimal conflict also requires innovation and creativity to build schemes that actively involve all stakeholders. The potential for friction between stakeholders certainly requires improvement in the government's ability to strengthen synergy, coordination, and dialogue between government agencies [321] and all stakeholders in various forest management schemes.

Policy and regulations are also essential factors that need to be addressed, for example, by eliminating ambiguity and asynchronous laws that overlap between the center and regions, both provinces and districts. It is also necessary to strengthen regulations that guarantee sustainable management by the community and strengthen the integration of FMUs in social forestry regulations [121]. In addition, the government seemingly needs to cut the time required to issue forest management legal permits [323,324] and simplify bureaucracy [121].

### 4.6. Biodiversity Hotspots vs. Deforestation

As one of the seventeen "megadiverse" countries, with two of the world's twenty-five "hotspots", eighteen "Global 200" ecoregions, and twenty-four "endemic bird areas" [325,326], Indonesia faces many challenges because it is located in the tropics and has many biodiversity hotspots that are also potential production areas. This condition poses a risk of conflict and becomes a major challenge in balancing the interests of conservation and food security. Referring to the food security and biodiversity conflict risk index developed by Molotoks et al. [327], Indonesia is one of the countries with the highest conflict risk in the world. If the biodiversity hotspot is overlaid with the Global Food Security Index (GFSI), then Indonesia is classified as a country with a biodiversity hotspot with a low GFSI value and is under great pressure due to food security.

There are two biodiversity hotspots in Indonesia, namely the Sundaland hotspot, which includes Kalimantan, Java, and Sumatra, as well as the Wallacea hotspot covering Eastern Indonesia [56,328,329]. The key drivers of biodiversity loss in all ASEAN countries are climate change, habitat change, invasive alien species, overexploitation, pollution, and poverty [330]. The main threats to biodiversity loss are decentralization, legal and illegal logging, oil palm plantations, wildlife trade and poaching, road building, mining, and civil conflict [331].

In Southeast Asia, Borneo Island is one of the hotspots with the highest diversity of flora and fauna after Indochina [332]. Borneo has diverse sites, habitats, and landscapes. On this third-largest island on Earth, everything from charismatic orangutans to the magnificent flower of Rafflesia can be found. The habitat is complete, from the mangrove forest at sea level to the mountain ecosystem on the peak of Kinabalu [333]. In order to conserve one of the remaining rainforests and water catchments in the interior of Borneo, Indonesia, Malaysia, and Brunei Darussalam initiated a campaign called The Heart of Borneo (HoB). HoB covers various areas in Indonesia (West, Central, East, and North Kalimantan), Brunei Darussalam (Belait, Temburong, and Tutong), and Malaysia (Sabah and Sarawak). The HoB area in Indonesia is approximately 16,835,379.44 ha and includes 17 districts [334].

Deforestation and conversion to other land uses are threats to biodiversity. Globally, it is a phenomenon experienced by many countries in the world, but in different phases, and Southeast Asia has the highest rate [335,336]. Indonesia is more vulnerable to deforestation than any other Southeast Asian country due to a high number of threatened species and invasive alien species [56]. The most important cause of deforestation in Indonesia was forest conversion for agricultural land and other development activities [337]. This condition poses a risk of conflict and becomes a major challenge in balancing the interests of conservation and food security.

The highest forest loss between 2000 and 2010 occurred in Kalimantan, Sumatra, Papua, Sulawesi, and the Moluccas due to fiber plantations and logging concessions [338], followed by large-scale oil palm establishments and timber plantations [339]. Indonesia's deforestation rate remained among the top five in the world until 2020 [214]. However, its rate tends to decline over the last five-year period (2015–2020), i.e., 425,309.02 ha/year. During the monitoring period 1990–2021, the highest rate of deforestation in 1996–2000 (±3.5 million hectares) decreased to 0.11 million hectares in 2020–2021 [1].

The commitment of Indonesia to halt deforestation has been shown in forest and environmental governance. Indonesia has enhanced peat governance through different regulatory mechanisms, notably the 3Rs strategy (rewetting, revegetation, and rehabilita-

tion of local livelihoods) and institutional components of peat management that focus on sustainability rather than just economics [340]. Moreover, community livelihoods must be considered in ecological restoration, and communities must have viable, suitable livelihood options. As for the HoB, the challenges for conservation include less biodiversity exploration and qualified experts to recognize the treasures of this area. Many areas of HoB are inaccessible for scientific exploration and expeditions. Recently, the wildlife trade for unique and ornamental species of plants and animals has become a threat, along with climate change's anomalous impact on species and ecosystem carrying capacity [341]. Many species are missing before adequate assessments of their taxonomy and/or conservation value are completed.

*4.7. Hydrometeorological Disaster and Forest Fire*

The number and frequency of natural disasters in Indonesia have a tendency to increase. Between 2005 and 2020, there has been a gradual increase in hydrometeorological disasters based on data generated by the Indonesian Disaster Data and Information (https://dibi.bnpb.go.id/, accessed on 3 January 2023). In terms of global warming, Indonesia is projected to have higher temperatures and changes in rainfall patterns in both the rainy and dry seasons. In terms of global warming, Indonesia is projected to have higher temperatures and changes in rainfall patterns in both the rainy and dry seasons. This situation poses a threat to floods and/or landslides during the rainy season and drought and/or fire during the dry season.

4.7.1. Hydrometeorological Disaster

Hydro-meteorological disaster is a term for natural disaster phenomena or destructive processes that occur related to the atmosphere (meteorology), water (hydrology), or oceans (oceanography) [342], including extreme weather, floods, landslides, drought, and land and forest fires [343,344]. Extreme weather and its derivative disasters can cause a range of harmful impacts, including loss of life, damage to infrastructure, economic losses, environmental damage, and public health impacts. They are the most frequent disasters in Indonesia [345], resulting in an annual material loss of an estimated IDR 33 trillion, or around USD 2.3 billion [346]. Based on BNPB data and information (https://dibi.bnpb.go.id 3 January 2023), the number of victims caused by extreme weather events and their derivative disasters during the 2018–2022 period reached approximately 25 million, including the dead, missing, suffering, injured, and displaced. Meanwhile, the damage to infrastructure, such as facilities for education, health, offices, and economics, as well as houses and bridges, reached about 250 thousand. Various studies have stated that the escalation in hydro-meteorological disaster events is related to a global climate change phenomenon that has hit the world in the last few decades [347–349]. Numerous studies have found that the upstream watershed's land cover change from forest to non-forest has increased river discharge, which has an effect on the likelihood of flooding downstream [350]. The hydro-meteorological disaster also causes losses and damages that impact the economy. Indonesia is ranked 13th in the world and has suffered economic losses and damages due to disasters [351]. Since hydro-meteorological disaster events continued to increase and resulted in huge losses due to catastrophes, it requires a response in disaster management through hydro-meteorological Disaster Risk Reduction (DRR), including institutional and governance (Figure 2).

4.7.2. Land and Forest Fire

Land and forest fires occurred in ENSO (the El Niño-Southern Oscillation) years, namely 1972, 1982–1983, 1987, 1991, 1994, 1997–1998, 2015, and 2019 [352,353]. Forest fires in Indonesia were mostly triggered by land clearing activities for agriculture and forest plantations [354,355], transmigration/resettlement schemes [356], and oil palm plantations [353,356].

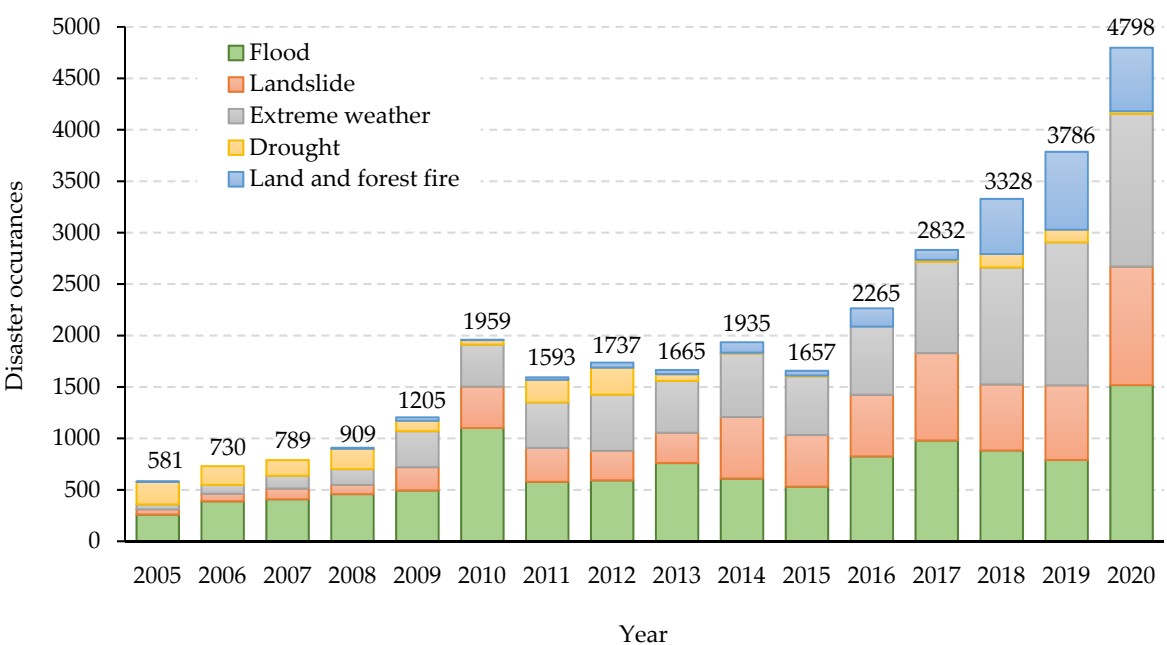

**Figure 2.** The number of hydrometeorological disaster events in Indonesia from 2005 to 2020 (extracted from https://dibi.bnpb.go.id/ accessed on 3 January 2023).

The severe fires not only had a bad impact on environmental degradation in the region [357] but also caused the spread of smoke to cause dense haze in a number of neighboring countries [352] and significant adverse impacts on ecosystems, economies, communities, and climate regionally and globally [358]. Based on the 1997/1998 forest fire study, the species richness and composition of butterflies, odonates, and plants were significantly different between unburned and burned forests [359]. Forest fires in 1997/1998, 2015, and 2019 resulted in economic losses of around USD 2.7 billion, USD 16.1 billion, and USD 5.2 billion [360,361]. The carbon emission from land and forest fires in 1997 was equivalent to between 13 and 40% of mean annual global carbon emissions from fossil fuels [362], while peat fires for four months only in 2015 reached 0.002 Gtonnes [363]. In fact, other hazardous gases and materials that reduced air quality were also significantly produced, such as $PM_{2.5}$ and carbon monoxide [364,365]. At the regional and local levels, fires cause changes in biomass stocks, alter the hydrological cycle, and impact the functioning of plant and animal species [354].

Forest fire prevention has become an emerging issue of global concern over the last three decades [356]. The government has made various efforts to deal with forest and land fires, both preventive and repressive [366]. A brigade tasked with controlling forest fires was formed by the Ministry of Forestry in 2003, named *Manggala Agni*, and spread over 37 operational areas in Indonesia [367]. The big fire accident in 2015 led the government to strengthen the institutions mandated to deal with climate change and fire issues: the Directorate General of Climate Change Control, an agency under the Ministry of Environment and Forestry [368], and the Peatland Restoration Agency, which is the arranger and facilitator for the restoration of 2 million hectares of peatlands in several provinces over a period of 5 years [369]. In addition, several early warning systems and disaster mitigation technologies are being developed by scientists in an effort to reduce risk and loss of life [370–372].

4.7.3. Disaster Risk Reduction (DRR) Strategy

There has been a shifting paradigm in Indonesia's disaster management based on various disaster occurrences, experiences, and issues related to disaster management. Some significant changes include a shift from a responsive to a preventive perspective, the involvement of stakeholders from specific sectors to multi-sectoral, and from top-down

management to more participatory management. DRR can be carried out by mainstreaming it into sectoral development, especially the forestry sector, and integrating traditional knowledge and scientific knowledge [373].

The National Disaster Management Agency (BNPB) has issued Decree No. 2/2012 concerning general guidance for disaster risk assessment. The scope of disaster risk assessment includes assessments of the threat level, vulnerability levels, capacity level, level of disaster risk, and disaster management policies based on the results of studies and disaster risk maps. The problem is data availability as well as data quality [70]. Along with the development of technology and science, the use of GIS technology, RS, and modeling techniques are scientific approaches that can be used [374]. The utilization of remotely sensed data in the GIS framework could become the answer to data limitations. The utilization of GIS/RS technology is recognized as a more effective and efficient approach in terms of financing and is capable of handling various data sources and scales both spatially and temporally [70].

Another important aspect is mainstreaming traditional community knowledge of disaster preparedness, mitigation, and adaptation [375,376]. Various traditional seasonal calendars, such as *Sasih* in Bali, *Pranotomongso* in Central Java, and the seasonal calendar in *Keuneunong* Aceh, are forms of community documentation regarding the changing seasons for anticipating possible hazards [68,377].

It is also important to consider a community-based approach and integrate indigenous and scientific knowledge for DRR, as stated in Indonesian National Law No. 24 /2007 on Disaster Management. Integrating local and indigenous knowledge with science and technology involves a participatory process that involves mobilizing community leaders, implementing awareness-raising activities, strengthening local organizations, and establishing linkages with local government.

Another related issue in hydro-meteorological DRR is ecosystem-based DRR. Conventional hydrometeorological disaster mitigation through the construction of embankments, dams, river channels, and artificial drainage systems often has a negative impact on the ecosystem [349,378]. The ecosystem-based solution demands a more detailed analysis of ecosystems' exposure, vulnerability, and resilience, as well as the interactions between social and ecological systems through the provision of ecosystem services. The principle is to reduce risk by modifying disaster characteristics and reducing the exposure and vulnerability of social-ecological systems. Therefore, balancing social and ecological considerations is critical in assessing existing vulnerabilities and risks [349]. All hydro-meteorological disasters cannot be entirely reduced by ecosystem-based DRR due to different geographic conditions, including social-economic dynamics. The hybrid strategy of integrating ecosystem-based and structural engineering can be the solution.

In Indonesia, the integration of DRR into forest management could be performed in several ways. These schemes include reforestation/afforestation programs, wildlife habitat development, tree seed development, rehabilitation of degraded land, management and development of national parks, and social forestry [68,379]. The key aspects could be increasing forest cover, protecting endangered ecosystems and species, developing tree seeds that are resilient to climate change, promoting community and social forestry, and promoting public awareness of the community.

*4.8. National Policies on Forest and Land Reformation*

Deforestation and forest degradation are major concerns for many developing countries, including Indonesia, and are related to the demography and dynamics of communities around the forest. These conditions became the baseline for the government to improve forest management policies, especially in Java, which has a high population. During Indonesia's independence, Perum Perhutani, a state-owned forestry company, largely managed forest management. So far, there has been no policy breakthrough to address the high level of community problems in the forest area.

MoEF has a big task: improving the welfare of the local community within and around the forest area through some scenarios. The first scenario is by providing objected land for agrarian reform (*Tanah Objek Reforma Agraria*/TORA) from the total forest area of at least 4.1 million ha. Meanwhile, the second scenario is to increase public access to forest management through social forestry on at least 12.7 million hectares [380]. The third is Forest Areas with Specific Management (*Kawasan Hutan dengan Pengelolaan Khusus*/KHDPK). KHDPK are designated forest areas from protected forests and production forests in which management is not delegated to state-owned forestry companies (Perhutani). The KHDPK policy is implemented in an area of 1.1 million hectares, which has been managed by Perhutani [381].

4.8.1. Objected Land of Agrarian Reform (TORA)

Although the percentage of the state forest area is still above the permitted threshold of 30%, in some areas, such as Java, Lampung, and Bali, the extent of the forest has not exceeded this specified threshold [380]. As mandated in the 2015–2019 RPJMN, the MoEF is tasked with identifying forest areas to be released for TORA of at least 4.1 million ha [38]. However, TORA implementation has only reached 2.6 million hectares until 2019, or about 63% of the target [382], so the 2020–2024 TORA period must complete the target of 1.5 million hectares to ensure this activity is fully implemented [38]. TORA's noble goals include reducing inequality in land tenure and ownership, creating sources of agrarian-based prosperity and welfare for the community, creating jobs to reduce poverty, improving community access to economic resources, increasing food security and sovereignty, and improving and maintaining environmental quality, as well as handling and resolving agrarian conflicts [383]. As a result, the successful implementation of the TORA program is part of the government's efforts to reorganize assets, specifically the realignment of control, ownership, use, and utilization of land, in order to create justice in the field of land tenure and ownership. In this case, the MoEF is quite progressive in following up on government policies regarding TORA. As a result, subjects who have received TORA must be included in community empowerment programs based on land use, including capital assistance, technical assistance, and access to other economic sources, until they are ready to be independent.

The procedures and requirements for releasing forest areas are regulated in MoEF Decree SK. 180/Menlhk/Setjen/kum.1/4/2017 and relate to problems that arise from agrarian reform (uncertainty in tenure or land ownership) [384]. Meanwhile, if there is a dispute over ownership in the area, Presidential Regulation Number 88 of 2017 concerning the Settlement of Land Tenure in Forest Areas must be referred to. Thus, the role and synergy (especially) between the Ministry of ATR/BPN and the MoEF are important in the TORA program, but caution is needed in deciding it, given that the area of Indonesia's forests is growing over time.

The location of TORA is determined based on four principles that have been created but are not applied to provinces with forest areas below 30% [380]:

a. Assignation of 20% of the area for community plantation land, which originates from enclave forest that had been released previously for large plantations according to Forestry Minister Regulation Number P.17/Menhut-II/2011, with a potential area of 1,254,705 ha.

b. Completion of the process of releasing forest area for transmigration settlements that have received principle approval from the Minister of Environment and Forestry, with a potential area of around 514,269 ha.

c. Deliverance of residential areas, public facilities, social facilities, and community agricultural land in forest areas with a potential area of about 2,582,981 ha.

d. Release production forests for food reserves in Central Kalimantan, West Kalimantan, and East Kalimantan, with a potential area of around 67,085 ha.

The high challenge of realizing this program requires a detailed spatial plan accompanied by field validation and verification. The role of local governments and stakeholders is

needed. Wise redistribution needs to be conducted to prevent the emergence of new social conflicts. Land inventory, which is currently dominated by the community, needs to be conducted at the village level. The relative positions of the claimed land conform to the TORA maps that have been prepared [380].

4.8.2. Forest Area with Special Management (KHDPK)

Forest area with special management (KHDPK) is a new terminology and scheme for the management of production and protected state forest areas in Java based on Government Regulation No. 23/2022 [1,385]. This program is implemented to improve Perhutani governance and Java forest governance. This program separates the management of non-conservation forests in Java, which was originally managed entirely by *Perhutani,* from being partly managed directly by the government (c.q. MoEF). Directions for the utilization of forest areas designated as KHDPK covering an area of 1,103,941 ha accommodate at least six interests, namely (a) social forestry; (b) area arrangement in the framework of forest area gazettement; (c) use of forest areas; (d) forest rehabilitation; (e) forest protection; or (f) utilization of environmental services [1,385].

The government's enthusiasm for realizing legal access for the community to utilize and manage forests that are accommodated by the broad social forestry program in the KHDPK should be appreciated, although it still needs further elaboration to improve it. The program of social forestry in Perhutani (*Izin Pemanfaatan Hutan Perhutanan Sosial/IPHPS*), launched by the MoEF in 2017, faced many problems at the site level, such as the IPHPS objects that were not suitable because they used forest areas with more than 10% coverage, and assistance to the community was not optimal [381]. Even though, since 1995, Perhutani has implemented CBFM, which makes the community and stakeholders partners to manage forests, in IPHPS, the community is the main actor in forest management [386]. Therefore, harmonizing the social forestry programs in the context of KHDPK and the PHBM program is necessary to avoid conflicts at the site level.

The consideration for the assignment of non-conservation forest management in Java from the government to Perhutani was changed based on Government Regulation No. 72/2010, which became Minister of Environment and Forestry Decree No. 73/2021 by stating a management area of 1,380,682 ha. On the other hand, if referring to Article 125 of Government Regulation No. 23/2021, the government's assignment of forest management to state-owned companies is regulated separately in a government regulation [385]. Therefore, clarity and consistency of considerations through accelerating the preparation of government regulations are urgently needed so that the KHDPK policy and the existence of Perhutani are not legally flawed. All efforts to improve the policy are solely to carry out the vision of "sustainable forests for a prosperous society".

*4.9. Global Agreements and National Commitments*

The issuance of Law No. 11/2020 concerning job creation can have impacts on climate mitigation efforts. This law aims to attract investments to create more jobs and enhance the country's economic development by simplifying policies and business processes, including in the forestry sector. Subsequent regulations of this law, Government Regulation No. 23/2021 concerning Forestry Management, mention that unproductive production forests can be converted for other purposes, such as food production. Even though these areas still have the potential to sequester carbon by protecting the remaining forest and rehabilitating the degraded areas, the regulation permits continuing cultivation under the *jangka benah* program in TORA and social forestry schemes, which may hinder the accelerated increase of carbon stock from forestry plants [1]. The Job Creation Law and Government Regulation No. 23/2021 also emphasize spatial planning control under the central government, which will influence local government decision-making regarding jurisdictional climate initiatives in the land sector [387]. Another issue in implementing the job creation law is related to fire prevention by eliminating strict liability, which can result in difficulties for future investigations against the responsible parties [388].

However, the job creation law also allows for more forestry-related businesses, such as carbon trading and offsets. This law encourages the implementation of multi-business forestry licenses in forest areas, enabling the private sector to participate in climate mitigation efforts. This is also supported by carbon pricing regulation, which was developed as an economic instrument to accelerate the achievement of NDC targets. Not only encouraging the private sector but also local and indigenous communities' participation and interests must be considered. Therefore, it is important to separate areas designated as results-based payment emission reduction programs, which are usually under the government to incentivize local communities, from areas for carbon trading or carbon offsets managed by the private sector [387]. In summary, the challenges presented here have also been found to be similar towards reaching sustainable forest management in the Amazon and Congo forests [389–395].

## 5. Strategic Recommendations: Toward a Viable, Sustainable Forest Management (Balancing Environmental, Social, and Economic Objectives)

### 5.1. Policy and Legal Aspects of SFM

#### 5.1.1. Devolution Policy

The basic concepts, goals, and forms of decentralization cover not only the transfer of power, authority, and responsibility in government but also the distribution of authority and resources to shape public policies in society. In Indonesia, this strong desire emerged in line with political developments and development growth in regions throughout the country.

From the evaluation of the devolution process in resource management in the form of community-based management, not all have been successful. There is a complexity of factors in each resource management case that influence the outcome. Effective devolution takes time, requiring a shift from a static concept of governance to a dynamic concept built on a process of feedback interaction that is adaptive over time [396].

#### 5.1.2. Evidence-Based Policy (EBP)

EBP is a set of rational, rigorous, and systematic policy formulation process methods that are not intended to directly affect the ultimate goal of the policy. The basis for the development of EBP is the premise that good policy decisions must be made based on evidence and rational analysis [397]. The concept of evidence-based policymaking has received attention in recent years. The use of sound evidence can make a difference in policymaking in at least five ways [398]: (a) achieve recognition of a policy issue, (b) inform the design and choice of policy, (c) forecast the future, (d) monitor policy implementation, and (e) evaluate policy impact.

In Indonesia, the use of evidence as a new paradigm in policymaking is currently considered very important and opens up great opportunities for researchers to be actively involved in policymaking through collaboration with policymakers. However, efforts are needed to ensure that research findings are accessible and translated into practical policies by policymakers. Not all research is suitable as a source of evidence for the policy-making process due to several factors, including incomplete research due to budget constraints, discrepancies between research directions and policy needs, and a research scope that is too small and not operational [6].

An evidence-based policy can play a role in the policy cycle, from determining the policy agenda and alternative choices to executing policies and monitoring the impact and output of a policy. In order for policies to be considered evidence-based, they must be evaluated using empirical and scientific approaches to determine whether they have a better or worse impact than the control group. If it is better, the program can be replicated for more people. EBP emphasizes the importance of getting the facts right.

*5.2. Implementation Approach Aspects*

5.2.1. Collaboration, Participation, and Community Involvement

Approaches to involving local people in forestry have multiplied over the years. The terms "social forestry", "community forestry", "rural development forestry", "joint forest management" (JFM), "shared forest management", "co-management", and "participatory forest management" are among those mentioned. Each of these has a distinct meaning and is frequently connected to specific projects or initiatives. All are essential interventions, based to a varying extent on local people's knowledge and wishes, but "legitimized and strengthened" [399].

Collaboration in national forest management often takes place through community-based collaborative groups, which are local groups that come together at the community scale to address natural resource management issues associated with public lands and resources that affect the environmental or economic health of the community [400].

Furthermore, the participation of communities in forest management in Indonesia is usually associated with social forestry [401]. However, the communities need to receive clear information and consent to the program. There are seven levels of participation, from manipulative participation to independent movement [402]. A study on participatory approaches in two peat swamp forests in Central Kalimantan has shown that there was a different level of community participation. Kalawa Forest Village has functional participation, which is a bottom-up process, while the communities surrounding Sebangau National Park have collaborative participation [403].

Although there are many different kinds of partnerships that result from collaboration, true partnerships demand some degree of stakeholder equity in decision-making. Although partnerships formed through collaboration can take many different forms, true partnerships call for some equity in how decisions regarding managing forests are made by various interest groups [399]. This component of cooperation is especially crucial now because poverty alleviation, improved governance, and social change are all possible through people-centered forestry, which has been implemented in Indonesia in the social forestry program [23].

Nowadays, a collaboration of various stakeholders is called Penta-helix. The key to Penta-helix success is a good engagement process that involves a wide range of stakeholders from each of the framework's four pillars: the public, private, academic, NGO, and civic society. In Indonesia, the penta-helix model has been implemented in the tourism industry of Borobudur temple [404] and in building e-commerce for coffee agroforestry in West Java [405]. She reported that, up to date, a penta-helix collaboration model had not been well developed in the forestry sector. Apparently, the penta-helix issue in SFM in Indonesia has not been well reported, although the implementation of penta-helix is emerging, owing to the extensive program of social forestry and the development of a business group of social forestry (*kelompok usaha perhutanan social*, *KUPS*) [406]. Further study is needed for the implementation of penta-helix collaboration in sustainable forestry management in Indonesia.

5.2.2. Financing for Sustainable Forest Management

Forest and environmental management are under the authority of the central government and local governments. To implement this authority, it must be supported by funding sources. This is regulated in the system of fiscal transfers to the regions and village funds and fiscal transfers at the provincial government level. Fiscal transfers to regions and village funds related to environment and forestry are made through balancing fund instruments consisting of the general fund allocation (*Dana Alokasi Umum*, DAU), profit sharing fund (*dana bagi Hasil*, DBH), special fund allocation (*Dana Alokasi Khusus*, DAK), regional instrument fund (*Dana Insentif daerah*, DID), and village fund (*Dana Desa*, DD) [407,408]. However, the existing fiscal transfer instruments are still general in nature, even though they are already ecologically based but have not specifically adhered to ecologically based fiscal principles [409]. At the regional level, ecologically based provincial budget transfers

have been implemented in several provinces, which are regulated by governor regulations, and at the district level, they are regulated by recent regulations.

One of the objectives of this fiscal transfer is to reduce the fiscal gap between the central government and local governments, but in practice, there is still a fiscal gap. Areas with large forest cover tend to have low fiscal capacity compared with areas with little forest cover [410]. Supposedly, areas with large forest cover receive incentives because they have taken good care of their environment. Areas with large forests require high fiscal requirements for the protection and preservation of biodiversity and the environment, so sustainable funding is needed by considering ecological aspects [411].

Sources of sustainable funding can come from within the country in the form of public and non-public funds and from abroad in the form of public and non-public funds. The environmental funding model consists of pure state budget funding, commercial, philanthropic, government, and business entity partnerships, blended finance, and impact investment [411]. Regulations for financing forest and environmental management in Indonesia are presented in Table 2.

**Table 2.** Regulations for financing forest and environmental management in Indonesia.

| Regulations | Main Content |
| --- | --- |
| UU 32/2009 and Government Regulation No. 46/2017 | Mandate to develop environmental and economic instruments consisting of development planning and economic activities, environmental funding, incentives, and/or disincentives. Central and local governments are required to allocate budgets for the environment. |
| Government Regulation No. 23 Tahun 2021 | Mandate to the Central Government, Regional Government, and other parties to be able to provide incentives to parties who can restore, maintain, and/or preserve forests inside and outside forest areas. |
| Minister of Finance Regulation No. 50/2017 | Transfers to regions consist of: (i) Dana Perimbangan (Transfer Umum dan Khusus); (ii) Dana Insentif Daerah (DID); (iii) Dana Otonomi Khusus; and (iv) Dana Keistimewaan DIY. |
| Government Regulation No. 47/2015 | Provisions for village fund allocation. |
| Government Regulation No. 12/2019 | Management of financial assistance from the province and the regency. |

Source: Nurfatriani [411].

5.2.3. Payment for Environmental Services (PES) in Sustainable Forests Management (SFM)

A PES scheme has been known worldwide as a holistic tool for SFM [412]. There are three paradigms in the PES scheme in Indonesia: commoditized environmental services (CES), compensation for opportunities skipped (COS), and co-investment in environmental stewardship (CIS) [413]. Among those three paradigms, CIS techniques have the most potential to be pro-poor, as CES and COS assume property rights, which the rural poor frequently do not. CIS will frequently be a component of a multiscale strategy for the regeneration and survival of natural capital.

The SFM is designed to increase incomes and conserve forests by influencing production and reduction techniques, such as reduced-impact logging. Although economic incentives such as those built into systems of forest concessions can also be used, technical changes in output are the key tool [414].

There are four types of PES that were prevailing in Indonesia, such as (i) carbon sequestration and storage, (ii) biodiversity protection, (iii) watershed protection, and (iv) landscape beauty for tourism [415]. The evidence of economic benefit from PES varies among sites. For instance, providing rewards in the form of long-term tenure security for local farmers in Indonesia has positively impacted the households' livelihoods in Lampung [416] and biodiversity conservation in Cidanau [417].

## 6. Conclusions

The history of forestry management in Indonesia has evolved through a long process, especially related to contestation over the control of natural resources and supporting policies and regulations. Nonetheless, the dynamics of this process lead to the sharing of benefits in managing natural resources and the awareness that preserving natural resources is a top priority. This ensures that the benefits we enjoy now do not reduce the rights and opportunities for future generations to also receive them.

After more than four decades, from the New Order regime to the reformation period, forests have become the government's major source of income. In 2014, there was a fundamental change in forest management with the merger of the Ministry of Forestry and the Ministry of Environment. The Indonesian government takes the political position that forests are in critical condition, so their sustainability must be maintained. Forest utilization should not only be oriented towards timber. This government policy direction is very clear from the construction of the Ministry of Environment and Forestry, when environmental pollution, climate change control, sustainable management of production forests, social forestry, and environmental partnerships became the focus of programs at the Ministry of Environment and Forestry. Until now, the government has been struggling to realize forest management that leads to the realization of environmental sustainability and people's welfare through structured efforts starting from planning to implementation. Forestry planning aspects have included efforts to support sustainable forest management. Strengthening coordination and integration with other sectors in forest planning is expected to reduce obstacles and conflicts during field implementation. Nature conservation policies continue to develop dynamically for the long-term sustainability of Indonesia's unique biodiversity.

The development of social forestry (SF) is a national priority. The SF target area has been drastically expanded from the initial target, from less than 1% (1.1 million hectares) to more than 10% (12.74 million hectares) of the forest area, to improve community welfare, environmental balance, and socio-cultural dynamics. However, despite its achievements related to the main issues of forest management in Indonesia, i.e., poverty, inequality in forest access, and deforestation, there are some challenges in the SF implementation, including tenure conflict incidents, a lack of consideration for biodiversity support capacity, an insufficient number and capacity of extension workers, and limited social forestry assistance.

Regarding human resource development, the government relies on forestry high school graduates and graduates from forestry, agriculture, geography, and other relevant disciplines. From the technical knowledge aspect, the government is revitalizing the extension system, vocational schools, and training; forestry community development, business-based training, and apprenticeship; increasing awareness and knowledge of the younger generation; and realizing good forest governance.

Even though the Ministry of Environment and Forestry was built to learn from past failures and improve forest governance, conflict is unavoidable. Conflicts between economic interests and the interests of environmental preservation, as well as between actors, are still serious homework. This can be minimized by carrying out responsible forest management through strengthening aspects of forest governance, eliminating ambiguity and overlapping regulations, overcoming conflicting interests proportionally, and simplifying the bureaucracy. In addition, the value of local wisdom must be maintained. Indigenous peoples and traditional village institutions in forest management must be strengthened to increase benefits for all stakeholders and involve them in the process from upstream to downstream.

Law enforcement is an important and integral part of sustainable forest management. The main barriers of law enforcement implementation come from structural (lack of coordination among law enforcement institutions at central and local levels), substantial (regulation on forest land use overlapped in its regulation among other regulations), and cultural (focusing on legal certainty and neglecting fairness and benefits aspects) sides. As a result, creating sound regulations for the environment, building an efficient institutional

system of law enforcers, encouraging active community engagement, and putting in place an integrated criminal justice system are all crucial for the effective implementation of law enforcement.

**Author Contributions:** All authors had an equal role as main contributors in discussing the conceptual ideas and the outline, providing critical feedback for each section, and writing the manuscript. All authors have read and agreed to the published version of the manuscript.

**Funding:** The English editing cost and article processing charges were supported by Yayasan Sarana Wana Jaya.

**Data Availability Statement:** Not applicable.

**Acknowledgments:** The authors thank the anonymous reviewers for their valuable comments. We are thankful for the financial support provided by Yayasan Sarana Wana Jaya to cover English editing costs and article processing charges. Disclaimer: The opinions and arguments in this paper are solely those of the authors and do not reflect the views of Yayasan Sarana Wana Jaya or the National Research and Innovation Agency (BRIN).

**Conflicts of Interest:** The authors declare no conflict of interest.

## Appendix A

**Table A1.** Key forest regulations by year and their implication on forests.

| Year | Regulations | Implication on Forest |
|---|---|---|
| 17th century to 1945 (Colonial period) | - The first Forestry Law (1865)<br>- The first Agrarian Regulation (*domeinverklaring*) in 1870 | - Legal basis for the Dutch colonial state to control land through the doctrine of *domeinverklaring*.<br>- All land without private entitlement is the domain of the state, formally categorizing it as "state-owned land." |
| 1945–1966 (Post-Colonial-Old Order Era) | The Basic Agrarian Law (BAL) No. 5/1960 | - Cancelled the *domeinverklaring* doctrine.<br>- The state has the power to control lands and resources (known as the "state's right" to control all lands and resources).<br>- On the one hand, the law recognizes the existence of customary rights and authority, yet on the other, it vests full power and authority with the state.<br>- The government declared 70% of its territory a forest zone without any reference to existing local tenure and land-use systems. |
| 1966–1998 (New Order Regime) | Basic Forestry Law No. 5/1967 | - This law defines two categories of forests: state-owned forests and privately owned forests, and it does not recognize customary territories. |
| 1970 | Government Regulation No. 21/1970 Government Regulation No. 33/1970 on Forest Planning | - Hundreds of concessions were quickly awarded through a discretionary non-bidding process to private, governmental, and para-statal firms for 20 years following the initial agreement.<br>- The forest area in Indonesia based on TGHK is 141,774,427 ha (Statistics of Indonesian Forestry 1990/1991), or +73.55% of the land area of Indonesia. |
| 1978 | State Forest licenses by the Ministry of Forestry | - Licensed state forest lands to both private and state-owned logging companies as well as to industrial timber plantation companies. |
| 1990 | Concession licenses by the government forest authorities | - The government forest authorities had granted concession licenses to more than 500 companies throughout Indonesia. It is estimated that about 60 million ha were offered for logging concessions for timber extraction, while over 4 million ha were granted for industrial timber plantations. |
| 1999 | Law No. 22 on Regional Governance | Provinces and districts received the authority to prepare their own rules, including forest management. |
| 1999 | Forestry Law No. 41 | - Cancelled the Basic Forestry Law No. 5/1967<br>- Forest authority was recentralized after being briefly delegated to districts.<br><br>The national government retains the right to:<br>- Organize and regulate all aspects of forests, forest estates, and forest products.<br>- Define the forest estate and/or update the forest estate's status.<br>- Define and regulate legal relationships between people and forests.<br>- Legal basis for social forestry schemes: forest management and utilization permits can be granted to individuals and cooperatives. |

**Table A1.** *Cont.*

| Year | Regulations | Implication on Forest |
|---|---|---|
| 2007 | Government Regulation 6/2007 on Forest Use and Forestry Management and Utilization Plan | Elaborates on the procedure for community plantation forests (HTR). |
| 2007 | Forestry Minister Regulation Number P.23/Menhut-II/2007 | On HTR. |
| 2011 | The Constitutional Court decision No. 45/PUUIX/2011 | - Gazettement of the forest estate/zone (*kawasan hutan*) is mandatory.<br>- Challenged the existing claims of the MOEF on *kawasan hutan,* covering 70% of the land area of Indonesia. |
| 2012 | The Constitutional Court decision No. 35/PUU-X/2012 | Defines that indigenous forests are private forests and not state forests. |
| 2014 | The Village Law No. 6 | Recognizes indigenous villages. |
| 2014 | Forestry Minister Regulation Number P.88/Menhut-II/2014 on Community Forestry | Revised HKm establishment processes, including the zoning of HKm areas, social mobilization, and facilitation by the government; it also sets the obligations to the communities. |
| 2014 | Forestry Minister Regulation Number P.89/Menhut-II/2014 on Village Forest | Establishment and obligations of the village forest zone, government facilitation, license granting, forest utilization, and logging permits. |
| 2014 | Joint regulation No. 79 on 'Procedures for the Resolution of Land Control in the Forest Zone | - The regulation was jointly issued by the Minister of Forestry, the Minister of Home Affairs, the Minister of Public Works, and the Head of the National Land Agency.<br>- The regulation grants land rights to the people who have been managing the land for over twenty years. |
| 2014 | Law No. 23 on Regional Governance | Shifts the power from the district to the provinces in relation to issuing permits for mining and logging. |
| 2015 | The National Medium Term Development Plan (2015–2019) | Allocates an area of 12.7 million ha for the local people, including customary communities. |
| 2015 | Ministerial Regulation No. 32 on title forest | - Title forest includes customary forests and forests on an individual/entity's title land.<br>- Defines the procedure to obtain a license for title forest: one can apply if it has a title from the National Land Agency, the local governments recognize the indigenous peoples and their ancestral land, or the Minister of Environment and Forestry recognizes the existence of title forest. |

Source: [30,418].

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
