# Peer review of "A Chronicle of Indonesia’s Forest Management: A Long Step towards Environmental Sustainability and Community Welfare"

_land, doi:10.3390/land12061238_

Round 1

Reviewer 1 Report

Land

A Chronicle of Indonesia's Forest Management: A Long Step 2 Towards Environmental Sustainability and Community Welfare

1. Don’ repeat abstract in introduction – abstract should focus on the new discoveries not a description of the context

2. Quality of English is quite good but still some copy editing is needed

3. I wonder if the journal has space for all the detailed regulatory text – this could be in supplementary material.

4. The text could be better balanced – it jumps from one historical period to another and gives a lot of attention to some issues – social forestry and climate change and much less attention to HPH – timber harvesting and biodiversity etc. 

5. Citation style is not consistent throughout – it even looks as though the paper is made up of text cut and pasted from independent reports and thus clashes of style have crept in. Many citations lack journal names and some simply give a date with not indication of what status the publication had.

6. There are some very interesting little facts scattered through the text which many readers will find interesting and which they might not find easily anywhere else.

7. The comprehensive bibliography is of great values and is worth publication in its own right – but it is a pity that many of the citations are not complete – they all need checking.

8. It would be good to check that none of the text is lifter verbatim from earlier publications – maybe using plagiarism detecting software – I’m not sure but wonder if some comes from technical reports from aid agencies etc.

9. This is not a well-balanced paper – it tries to do too much and is very uneven – a lot of detail on some things and huge gaps on other important things – like HTI expansion the successes and failures of HPH etc.

10. The paper does not have a clear objective – it consists of a long series of loosely connected pieces of text with not connection between them. It really needs some seriously heavy editing by someone who knows the subject very well. – I would like to see it published but I feel that it is not yet in a state what would make it acceptable. It is really a series of independent texts that do not add up to a single integrated paper.

11. Having given the matter further thought I have concluded that this submission does not really match the objectives and the editorial capacity of our journal. Perhaps the authors should be encouraged to look for an outlet that spécialisés in longer, descriptive accounts of governmental. Processes.

.

Reviewer 2 Report

The manuscript by Nugroho et al. (land-2361063) reviewed the long steps of Indonesian forest management toward achieving environmental sustainability and community welfare. This manuscript examined the historical evolution of forest management in Indonesia, displayed existing policies and strategy, presented various current and future challenges, and put forward strategic recommendations toward a viable, sustainable forest management. Overall, this manuscript is informative. I provide some comments below, which may be helpful for the authors to further improve the paper. I would suggest a minor revision of the paper before it is accepted.

Specific comment:

Please unify the indentation format of the first line of the paragraph. Line 146, 246, 266, and 280.

In section 4.7, the author exhibited the number and frequency of natural disasters in Indonesia have a tendency to increase, and focused on analyzing the hazards of forest fires. However, I would suggest the authors further describe the harm brought by the significant increased extreme weather events, because extreme weather events, like fire events, were also significantly increasing in figure 1.

Section 4 described the challenges of nature and humanity in the present and future in Indonesian forest management. It is appropriate to provide a comparison of the challenges of forest management in different regions (such as Congo forest and Amazon forest), which may help readers and decision-makers understand the current issues in forest management.

Table 2 “m3/yr should be m3/yr

The quality of the tables should be greatly improved. Please unify the format of the three tables
